# Structure-guided optimization of SLC1A1/EAAT3-selective inhibitors targeting renal cancer metabolism

Pooneh Koochaki[1,9], Biao Qiu [ID][2,3,9], Jesse A Coker [ID][4,5,9], Alexander Earsley [ID][2], Nancy S Wang[4], Todd Romigh[4], Christopher M Goins[4,5], Carleigh Salem[1], Dehui Mi[6], Emily Days[6], Joshua A Bauer[6], Shaun R Stauffer [ID][4,7 ✉], Olga Boudker [ID][2,3 ✉] & Abhishek A Chakraborty [ID][1,7,8 ✉]

## Abstract

Renal cell carcinomas (RCCs) depend on the trimeric sodium-coupled aspartate and glutamate transporter, SLC1A1/EAAT3; however, pharmacologically targeting SLC1A1 is challenging. Here we determined a cryo-EM structure of human SLC1A1 bound to compound 3e, a recently described SLC1A1-selective bicyclic imidazo[1,2 α]pyridine-3-amine (BIA) inhibitor with an unclear mechanism of action. 3e binds a membrane-embedded allosteric pocket accessible only in the apo state, when SLC1A1 is unbound to substrate and sodium, and likely prevents sodium and substrate binding. Moreover, by forming a wedge between the trimerization domain and the substrate-binding transport domain, alongside a cholesterol moiety from the lipid bilayer, 3e blocks SLC1A1's elevator-like movements that support the transport cycle. Mutations in this binding pocket abolish the 3e interaction and counteract 3e's cytotoxicity in RCC cells, confirming on-target activity and explaining SLC1A1 selectivity. The subsequent design of two new SLC1A1-selective BIA derivatives, PBJ1 and PBJ2, was directed by the SLC1A1-3e structures; both inhibited SLC1A1-dependent aspartate, glutamate, and cysteine metabolism and showed enhanced cytotoxicity.

Subject Categories Cancer; Structural Biology

## Introduction

Solute carriers (SLCs) are increasingly recognized for their roles in human diseases, including cancers (Hediger et al, 2013). Their localization to the cell surface makes them accessible to pharmacological agents, rendering them inherently druggable and therapeutically valuable. Furthermore, compared to normal tissues, the aberrant metabolic demands of tumors heighten their reliance on SLC activity, creating a therapeutic window for selective cancer targeting (Lin et al, 2015).

We previously identified SLC1A1 (also known as Excitatory Amino Acid Transporter 3, EAAT3) as an oncogenic dependency in clear cell Renal Cell Carcinomas (ccRCCs) (Grubb et al, 2025). ccRCCs are the most common form of adult kidney cancer and originate from the proximal tubule epithelial cells (Mitchell et al, 2018; Turajlic et al, 2018). Others have reported SLC1A1 as a metabolic vulnerability in fibrosarcoma (Yang et al, 2023), lung cancer (Guo et al, 2021; Wen et al, 2025), hematologic malignancies (Xiong et al, 2021), and brain tumors (Palos et al, 1996). SLC1A1 imports aspartate (L-Asp), glutamate (L-Glu), and, to a lesser extent, cysteine (L-Cys) from extracellular space (Bailey et al, 2011; Himi et al, 2003; Kanai and Hediger, 1992; Watts et al, 2014; Zerangue and Kavanaugh, 1996). L-Asp is essential for cancer cell growth because it provides the carbon skeleton for nucleotide biosynthesis (Sullivan et al, 2015). Moreover, L-Asp's poor cellular permeability and scarcity in hypoxic environments lead to increased dependence on its transporters in solid tumors (Birsoy et al, 2015; Garcia-Bermudez et al, 2018). L-Glu supports reductive carboxylation in hypoxic tumors and, along with L-Cys, serves as a precursor for glutathione, aiding the oxidative stress response (Chen et al, 2024; Gameiro et al, 2013; Ling et al, 2023).

The SLC1A1 paralog, SLC1A2 (EAAT2), is predominantly expressed in the central nervous system (CNS), where it constitutes 1–2% of total protein in the forebrain (Danbolt et al, 2016; Danbolt et al, 1992). It is crucial for mammalian glutamate-mediated synaptic transmission and survival. In contrast, although SLC1A1 is broadly expressed and controls L-Asp/L-Glu reabsorption in the normal kidney, homozygous *Slc1a1* loss is well tolerated in mice. These mice show elevated urinary excretion of L-Asp and L-Glu (aminoaciduria); however, SLC1A1 loss does not cause neurodegeneration unless exposed to hypoxia or oxidative stress (Aoyama

[1]Department of Cancer Sciences, Cleveland Clinic Research, Cleveland, OH, USA. [2]Department of Physiology & Biophysics, Weill Cornell Medicine, New York, NY, USA. [3]Howard Hughes Medical Institute, Weill Cornell Medicine, New York, NY, USA. [4]Center for Therapeutics Discovery, Cleveland Clinic Research, Cleveland, OH, USA. [5]Cleveland Clinic Lerner College of Medicine of Case Western Reserve University, Case Western Reserve University, Cleveland, OH, USA. [6]Vanderbilt Institute of Chemical Biology, Vanderbilt University, Nashville, TN, USA. [7]Department of Pharmacology, Case Western Reserve University, Cleveland, OH, USA. [8]Case Comprehensive Cancer Center, Cleveland, OH, USA. [9]These authors contributed equally: Pooneh Koochaki, Biao Qiu, Jesse A Coker. ✉E-mail: stauffs2@ccf.org; olb2003@med.cornell.edu; chakraa@ccf.org

et al, 2008; Aoyama et al, 2006; Peghini et al, 1997). Similarly, SLC1A1 inactivation in humans results in aminoaciduria and minor cognitive symptoms, with minimal additional manifestations and a normal life expectancy (Bailey et al, 2011). These observations suggest that global (and transient) SLC1A1 inhibition is likely safe in humans, supporting the development of selective inhibitors for cancer therapy.

All SLC1 transporters utilize an "elevator" mechanism for substrate uptake (Fig. 1A) (Canul-Tec et al, 2017; Canul-Tec et al, 2022; Kato et al, 2022; Qiu and Boudker, 2023, 2025; Qiu et al, 2021; Reyes et al, 2009). These homotrimeric carriers consist of a central trimerization or scaffold domain and a peripheral transport domain, which moves across the membrane from the outward-facing (OF) to the inward-facing (IF) position. In the OF conformation, the transport domains bind the substrate and co-transported Na$^+$ from the extracellular space, then translocate to the IF conformation to release them into the cytoplasm. During the return to the OF state, they translocate a K$^+$ ion from the cytoplasm to the outside, completing the cycle (Qiu and Boudker, 2023; Qiu et al, 2021; Zerangue and Kavanaugh, 1996).

Most current SLC1 inhibitors are competitive L-Asp-derived compounds, such as DL-threo-β-benzyloxyaspartic acid (TBOA) (Shimamoto et al, 1998), and its trifluoromethyl-benzoyl derivative (TFB-TBOA)(Shimamoto et al, 2004), which lack selectivity and drug-like properties necessary for transition into the clinic. A recent study, however, described a series of bicyclic imidazopyridine amine (BIA) SLC1 inhibitors, with compound 3e emerging as the best-in-class SLC1A1-selective agent (Wu et al, 2019). Notably, 3e shares no structural similarity with SLC1A1's canonical substrates, raising questions about its on-target engagement and mechanism of action. Moreover, although 3e inhibits cancer cell growth in vitro, it exhibits poor solubility and low kidney bioavailability (Grubb et al, 2025). Lastly, 3e contains an unsubstituted furan ring, making it vulnerable to hepatic clearance by the cytochrome P450 enzymes. Thus, 3e is not suitable for in vivo use and limits opportunities for future clinical translation.

We report a cryogenic electron microscopy (cryo-EM) structure of human SLC1A1 bound to compound 3e. Remarkably, 3e occupies an allosteric pocket between the Na$^+$-binding site and the substrate-binding gate, accessible only in SLC1A1's apo state, unbound to Na$^+$ and substrate. The structure reveals the transporter in an IF apo state where 3e and a cholesterol snuggle between the transport and scaffold domains. These findings suggest a novel inhibition mechanism in which 3e competes with Na$^+$ and substrate binding and likely traps the transporter in the IF conformation. Functional analysis reveals that F99 and T402 in SLC1A1 are key residues for 3e-induced cytotoxicity, with the non-conserved F99 residue likely conferring selectivity towards SLC1A1. Structure-guided modifications of the BIA scaffold led to the identification of PBJ1 and PBJ2, two SLC1A1-selective inhibitors with improved cancer cell-killing potency. Genetic and metabolic rescue assays confirm that PBJ1 and PBJ2's cytotoxicity is on-target and stems from the depletion of key L-Asp-, L-Glu-, and L-Cys-derived metabolites. These findings reveal a previously unrecognized mechanism of SLC1 inhibition, providing a foundation for the development of SLC1A1-targeted therapies.

## Results

### Cryo-EM structure of SLC1A1/EAAT3 with Compound 3e

To assess 3e's mechanism of inhibition, we imaged recombinantly expressed human SLC1A1 with mutated glycosylation sites (SLC1A1g; see Methods) in the presence of 200 mM NaCl and 100 μM 3e by cryo-EM. Particle alignment with imposed threefold symmetry yielded an EM map at an overall resolution of 2.46 Å, revealing the transporter in an IF conformation (Appendix Fig. S1 and Appendix Table S1), similar to our previously reported structure (Qiu and Boudker, 2023; Qiu et al, 2021). Since each protomer in the trimer functions independently (Grewer et al, 2005; Koch et al, 2007; Leary et al, 2007; Qiu et al, 2021), we applied symmetry expansion and protomer classification without alignment, revealing three major structural classes (Appendix Fig. S1 and Appendix Table S1). One class, resolved at 2.83 Å, pictured an IF protomer bound to Na$^+$ ions (Fig. 1B), indistinguishable from the previously reported Na$^+$-bound SLC1A1g structure (Appendix Fig. S2A). Two other classes (designated A and B), resolved at 2.56 and 2.76 Å, respectively, resembled the apo IF conformation of SLC1A1g determined in the absence of Na$^+$ ions and substrates (Fig. 1C–E). No density corresponding to 3e was observed in the Na$^+$-bound protomers. In contrast, clear non-protein density consistent with 3e was present in both apo protomers (Fig. 1D,E), suggesting that 3e either selectively binds to apo SLC1A1g or displaces bound Na$^+$.

To confirm the direct binding of 3e to SLC1A1, we performed thermal shift binding assays using detergent-solubilized SLC1A1g. The protein was subjected to increasing temperature, and its unfolding was monitored using nano-differential scanning fluorimetry (nanoDSF). In a buffer containing 200 mM Na$^+$, SLC1A1g unfolded at ~69.1 ± 0.2 °C. The addition of 100 μM 3e produced a modest increase in the melting temperature of ~0.4 ± 0.3 °C (Fig. 1F; Appendix Fig. S2B). In contrast, in Na$^+$-free, apo conditions, 3e showed a more pronounced stabilization, with thermal shifts of ~1.8 ± 0.2 °C (Fig. 1F; Appendix Fig. S2C), suggesting direct binding. These findings are consistent with our earlier cellular thermal shift assays (CETSA), in which we detected 3e-induced SLC1A1 stabilization using extracts from human RCC cells expressing high levels of SLC1A1 (Grubb et al, 2025). Together with the structural data, these observations suggested that 3e competes with Na$^+$ for binding to the carrier. Consistent with this, we found that 3e led to a dose-dependent inhibition of Na$^+$ binding to the transporter, as measured by nanoDSF (Fig. 1G).

We further examined, using SLC1A1g heterologously expressed in HEK293 cells, whether inhibition by 3e depends on extracellular Na$^+$ concentration. IC$_{50}$ values were determined under saturating Na$^+$ concentrations (100 mM) or at a near-K$_m$ concentration (20 mM). 3e inhibited L-Asp uptake with IC$_{50}$ values of 4.3 ± 0.5 μM at 100 mM Na$^+$ and 1.8 ± 0.7 μM at 20 mM Na$^+$, respectively (Appendix Fig. S2D), indicating relatively weak Na$^+$ dependence. Altogether, this behavior is consistent with tight binding of 3e to the IF state and also likely reflects the relatively constant cytosolic Na$^+$ concentrations experienced by the transporter.

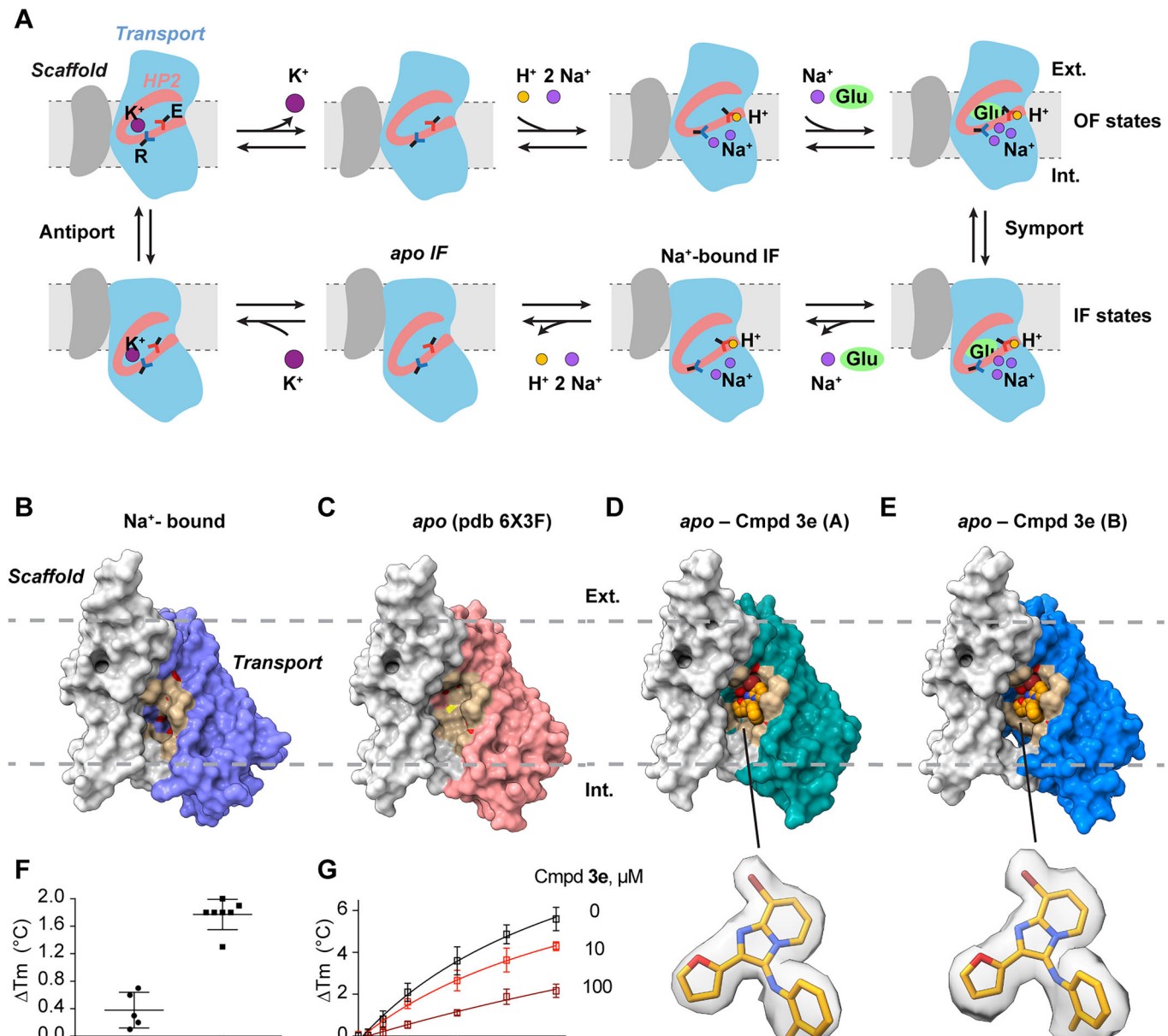

**Figure 1. Compound 3e binds to *apo* inward-facing SLC1A1g protomers.**

(A) The transport cycle of SLC1A1 transporters. Outward-facing (OF) carriers bind substrate and symported ions under occluding HP2 (raspberry); the transport domain (blue) translocates relative to the stationary scaffold domain (gray) into the inward-facing (IF) state, where it releases the substrate and ions; the transport domain returns into the OF state after binding a K[+] ion. Key ion and substrate-binding arginine (R) and glutamate (E) residues are shown for reference. (B, C) The surface representation of structures of Na[+]-bound SLC1A1g observed in the cryo-EM sample (B) and the *apo* transporter (C, PDB access code 6X3F) in IF. The structures show an enlarged drug-binding pocket in the *apo* carrier. The scaffold domain is colored white, the transport domain is in various colors, and the residues within 4 Å of 3e in the drug-bound states are colored in tan. The dotted gray lines represent the approximate boundaries of the membrane bilayer. (D, E) Structures of SLC1A1g bound to 3e as seen in the structural classes A (D) and B (E). The compound 3e is shown as spheres and colored orange. The EM density contoured at 5σ around the modeled 3e is shown below the panels. (F) Thermal stabilization (mean ± SD, $n = 5$) of SLC1A1g by 100 µM 3e in the presence and absence of 200 mM Na[+]. (G) Thermal shifts induced by increasing Na[+] concentrations, as indicated, in the presence of the indicated concentrations of compound 3e (mean ± SD, $n = 7$). The Tm measured at 0 mM NaCl was normalized to zero to demonstrate change in thermal stabilization. Source data are available online for this figure.

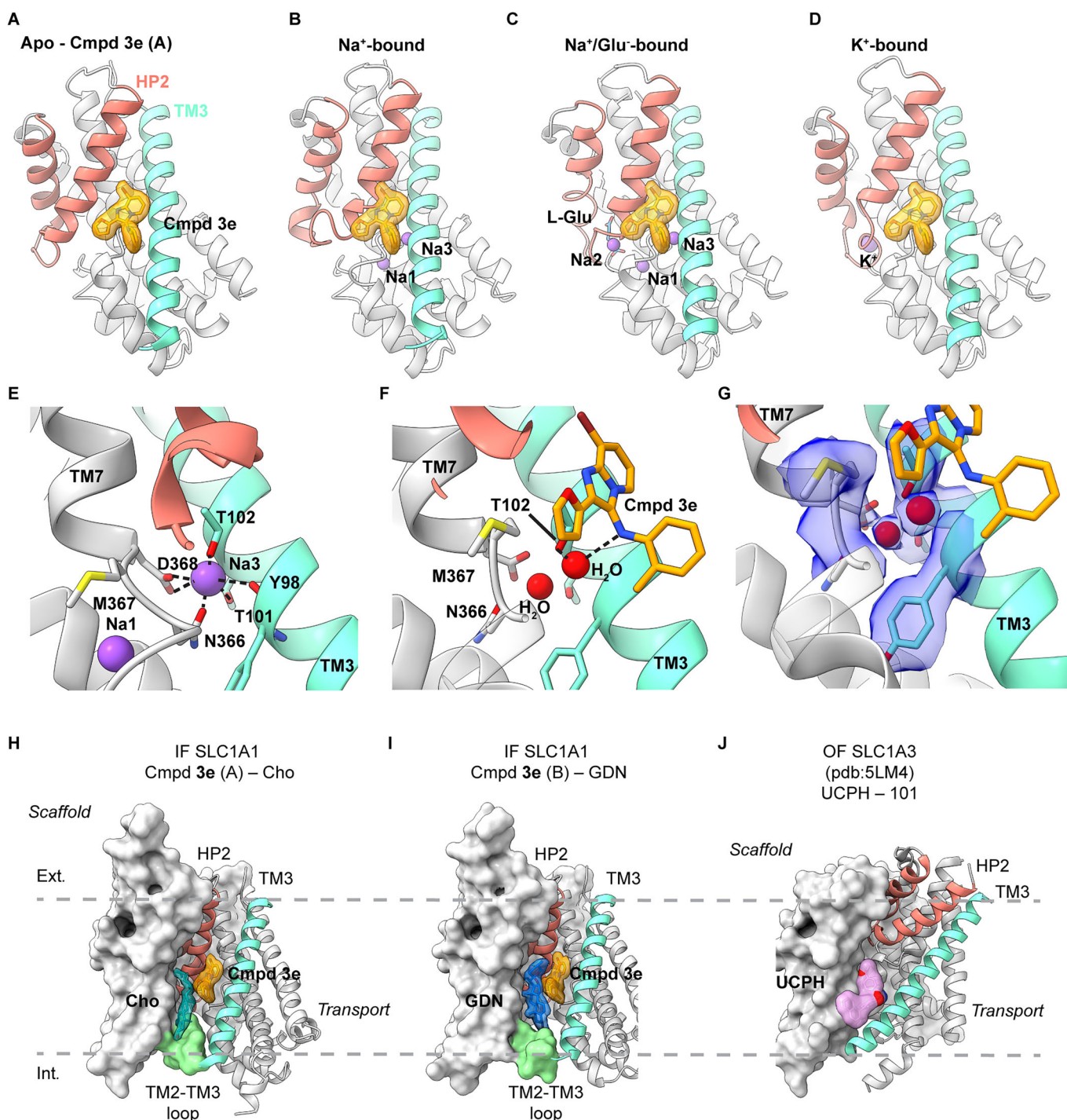

## Compound 3e and cholesterol bridge transport and scaffold domains

Closer inspection of the transport domain structure revealed that in both structural classes, A and B, **3e** binds within a pocket formed between transmembrane (TM) helix 3 (TM3) and the N-terminal helix of the helical hairpin 2 (HP2) (Fig. 2A). HP2 acts as a gate, controlling the access of substrates and ions to their binding sites (Alleva et al, 2020; Boudker et al, 2007; Focke et al, 2011; Riederer

et al, 2022; Verdon et al, 2014). HP2 closes upon binding of the substrate, three co-transported $Na^+$ (Na1-3) ions and a proton, or counter-transported $K^+$ ions (Fig. 1A), facilitating the elevator-like movement of the transport domain. Whereas the **3e** binding pocket is open in *apo* and $K^+$-bound states, HP2 shifts closer to TM3 and obstructs the pocket in the $Na^+$-bound and L-Glu-bound states (Fig. 2B-D). TM3 harbors the conserved T101 and T102 residues, which, together with N366 and D368 in TM7, coordinate bound $Na^+$ in the Na3 site (Fig. 2E). When bound, **3e** forms a water-

**Figure 2.  Compound 3e and Na⁺ ions compete for binding to SLC1A1g.**

(A–D) Structures of the *apo* SLC1A1g transport domain bound to **3e** (A). Modeled **3e** bound to SLC1A1g with Na⁺ only (B), Na⁺ and L-Glu (C), and with K⁺ (D). The transport domain is depicted as a gray cartoon, with HP2 and TM3 highlighted in raspberry and green-cyan, respectively. ʟ-Glu is depicted in blue, and Na⁺ and the K⁺ ions in purple. **3e** is shown as an orange surface where it is observed in cryo-EM (C) or modeled at corresponding positions (B–D). Modeling shows that in Na⁺-bound transporter, with or without Glu, the compound would sterically clash with HP2. The binding pocket is preserved in the K⁺-bound transporter. (E, F) Close-up view of the Na3 and **3e** binding sites in Na⁺- and **3e**-bound states of SLC1A1g, respectively. Residues coordinating Na3 are shown as sticks. T102 coordinates Na3 (E) and, via a water-mediated hydrogen bond, **3e** (F). (G) An EM density map contoured at 5σ around the modeled side chains and waters in **3e**-bound SLC1A1g in the B structural class. (H, I) Structures of SLC1A1g bound to **3e** with cholesterol in structural class A (Cho, H) and GDN in class B (I) sandwiched between the drug and the scaffold domain. Scaffold and transport domains are shown in surface and cartoon representations, respectively. The TM2-TM3 loop colored light green. The EM density around **3e**, cholesterol, and GDN is contoured at 5σ and colored orange, teal, and slate blue, respectively. (J) Structure of SLC1A3/EAAT1 in OF state bound to UCPH-101 (PDB code 5LM4). The drug is colored pink and shown in surface representation. Note that UCPH-101 is more inwardly positioned than **3e** and coordinated by the cytoplasmic end of TM3.

mediated hydrogen bond between its aniline NH and the hydroxyl group of SLC1A1's T102 (Fig. 2F,G). Thus, **3e** and Na⁺ in the Na3 site rely on the same T102 residue and favor different conformational states of the transporter.

In structural class A, we observed a well-resolved cholesterol molecule on the lipid-facing side of compound **3e** (Fig. 2H). Its headgroup was adjacent to the TM2-TM3 loop that links the scaffold and transport domains on the cytoplasmic side, while the hydrophobic sterol moiety was nestled between the drug and the scaffold domain. In the dominant class B, this site was occupied by glyco-diosgenin (GDN)—a detergent used during protein purification (Fig. 2I). Due to its larger size, GDN wedges more deeply into the crevice between the scaffold and transport domains, with its polar headgroup protruding beneath the TM2-TM3 loop. As a result, the transport domain swings slightly outward. In both structures, the hydrocarbon tails and parts of the sterol rings of cholesterol and GDN form intimate van der Waals contacts with **3e** on one side and the scaffold domain on the other. By contrast, no well-ordered lipid-like densities were observed at this site in the Na⁺-bound IF SLC1A1g structure.

Because **3e** binds within the transport domain, it remained unclear whether it might also engage the *apo* OF state or K⁺-bound conformations in which the **3e** binding pocket remains accessible (Fig. 2D). To address this, we used the K269C/W441C SLC1A1g mutant, which can be trapped in a mixture of the OF state and an intermediate between the OF and IF states by Hg²⁺-mediated cross-linking (Qiu and Boudker, 2023). Previously, we found that in the presence of KCl, the cross-linked mutant populates an *apo* OF state lacking bound K⁺ ions and a K⁺-bound intermediate state (Qiu and Boudker, 2023). Cryo-EM analysis of this mutant in the presence of 300 mM KCl and 100 μM **3e** resolved two structural classes corresponding to these states, refined to ~3.1 Å and ~3.2 Å resolution and comprising ~80% and ~20% of particles, respectively. Neither class exhibited well-defined densities attributable to **3e** or cholesterol at the expected binding sites (Appendix Fig. S3A,B). Although a weak non-protein density was observed at the **3e**-binding site in the intermediate state, which was absent in the previous dataset without compound, it could not be convincingly assigned to **3e**. These observations suggest that, in the presence of K⁺, **3e** either does not bind to this site with substantial occupancy and/or remains highly dynamic upon binding.

Our findings do not exclude the possibility that **3e** allows K⁺ binding to the *apo* IF transporter, a step expected to occur rapidly following substrate and Na⁺ release into the cytoplasm. We were unable to test this directly, as K⁺ binding to the IF SLC1A1g is weak —at least under the detergent conditions used for cryo-EM—and the K⁺-bound IF structure has not yet been resolved. Nonetheless, structural and functional data support a model in which **3e**, together with associated cholesterol, bridges the scaffold and transport domains, thereby impeding the elevator-like movements required for the transition from the IF state to the intermediate and OF conformations and effectively arresting the transport cycle. A conceptually related mechanism has been proposed for UCPH-101, an allosteric, non-competitive inhibitor selective for SLC1A3/ EAAT1 (Abrahamsen et al, 2013; Erichsen et al, 2010), which binds at the interface between the scaffold and transport domains in the OF state and prevents inward translocation of the transport domain (Fig. 2J) (Canul-Tec et al, 2017).

## Structural basis of 3e's SLC1A1 selectivity

Most residues forming the **3e**-binding pocket are conserved across SLC1 paralogs, except for F99, which is unique to SLC1A1, and which interacts with ortho-methylaniline of **3e** (Fig. 3A). To probe the role of the F99 residue, we mutated it to leucine or methionine —residues found at the equivalent position in other SLC1 paralogs (Fig. 3B)—or to alanine as a control. Additionally, we mutated the conserved T402 residue, which forms a hydrogen bond with the core imidazopyridine ring of **3e**, to isoleucine or leucine. We evaluated these mutants using cell survival assays to assess **3e**-mediated inhibition of SLC1A1 and nanoDSF to assess direct binding to the transporter.

We first generated, using site-directed mutagenesis, a panel of the SLC1A1 mutants (Table 1) and then tested their functionality in a rescue assay. CRISPR/Cas9-mediated knockout of SLC1A1 (1A1sg5) causes antiproliferative effects in human kidney cancer cells, such as A498, but viability can be restored by introducing a sgRNA-resistant version of functional SLC1A1. Except for the F99A mutant, all F99 and T402 variants retained functionality and restored cell viability to levels comparable to WT SLC1A1 (Fig. 3C). We then used isogenic A498 cells expressing either WT or mutant SLC1A1, or GFP as a non-specific control, in the presence of endogenous SLC1A1 and cultured them with various doses of **3e**. All functional **3e**-binding site mutants resisted the cytotoxic effects of **3e**, whereas cells expressing WT SLC1A1, the non-functional F99A mutant, or GFP remained sensitive (Fig. 3D). Notably, survival of the latter two cell lines depends on the endogenously expressed SLC1A1, which is inhibited by **3e**.

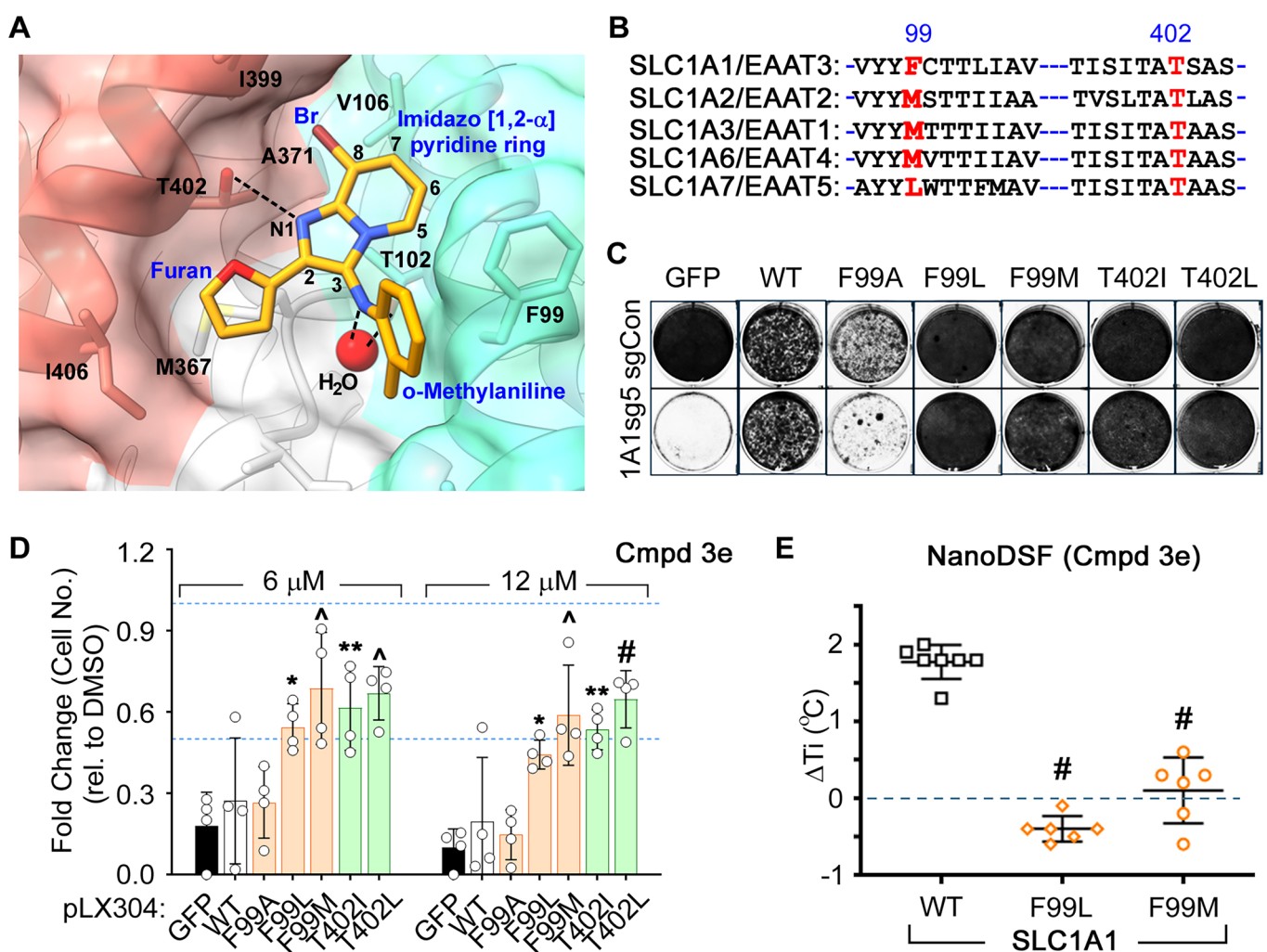

**Figure 3. Binding selectivity determinants and SLC1A1 drug-resistant mutants.**

(A) A close-up view of the **3e** binding pocket in cartoon and transparent surface representation. The SLC1A1 residues within 4 Å are shown as sticks and labeled in black. The **3e** backbone is numbered, and key functional groups are labeled in blue. Hydrogen bonds with the resolved water molecule (red) are shown as dashed lines. (B) Sequence alignment around TM3 residue F99 and HP2 residue T402 for SLC1 proteins. T402 is conserved while F99 is variable. (C) Crystal violet staining of A498 cells that were lentivirally transduced to express sgRNA (sg5)-resistant versions of either wild-type (WT) or mutant SLC1A1, or GFP (as a control), and then transduced with sgRNA constructs to inactivate SLC1A1 (1A1sg5) or a non-targeting control (sgCon). (D) Viable cell number of isogenic A498 cells expressing the indicated forms of SLC1A1 (or GFP control) and then cultured in the presence of the indicated concentrations of **3e** for 20 days. Cell numbers, relative to vehicle DMSO treatment (blue dashed line), were represented as mean ± SD, $n = 4$, and compared to the GFP arm using two-way ANOVA with Tukey's multiple comparison test. $P$ values at 6 µM—WT: 0.97, FA: 0.98, FL: 0.02[*], FM: 0.0002[***], TI: 0.0021[**], TL: 0.0004[*]. $P$ values at 12 µM—WT: 0.96, FA: 0.99, FL: 0.026[*], FM: 0.0004[*], TI: 0.0021[**], TL: <0.0001[#]. (E) NanoDSF results showing changes in $Ti$ for WT SLC1A1g versus the F99L and F99M mutants that lose the ability to bind **3e**. Experiments were repeated using two independent protein preparations with at least three technical repeats each and compared using one-way ANOVA with Dunnett's multiple comparison test, [#]$P < 0.0001$. Source data are available online for this figure.

We also introduced the F99L and F99M mutations into SLC1A1g. Both purified mutant proteins remained monodisperse in size-exclusion chromatography (Appendix Fig. S4A) and exhibited thermal stability similar to WT in the nanoDSF assays, with $Ti$ values of ~67.3 ± 0.5 °C (F99M) and ~65.6 ± 1.8 °C (F99L) (Appendix Fig. S4B,C). However, **3e** failed to stabilize these mutant proteins (Fig. 3E), consistent with substitutions at F99 disrupting **3e** binding. These results validate our cryo-EM structure and identify F99 as a key determinant of **3e**'s selectivity towards SLC1A1. Furthermore, these drug-resistant mutants confirm that **3e**'s cytotoxicity in kidney cancer cells is mediated through on-target inhibition of SLC1A1.

## Structure-activity relationship (SAR) studies to identify new SLC1A1 blockers

Compound **3e** binds within a lipid-embedded hydrophobic pocket and, unsurprisingly, is itself highly hydrophobic with a calculated octanol-water partition coefficient of 5.4 (cLogP). The ortho-methylaniline moiety orients toward the selectivity-defining residue F99, while the 2-furanyl ring is positioned beneath the adjacent cholesterol molecule, engaging in an arene-H interaction with the sidechain of M367 and packing against I406 (Figs. 2H and 3A). This binding pose explains **3e**'s previously reported empirical SAR

**Table 1.   List of Oligos used for Mutagenesis and Cloning**

| Mutagenesis primers | | |
|---|---|---|
| Name | Mutation | Sequence |
| 1A1F99Atop | F99A | GCTGTCGTGTATTATGCCTGTACCACTCTCATT |
| 1A1F99Abtm | F99A | AATGAGAGTGGTACAGGCATAATACACGACAGC |
| 1A1F99Mtop | F99M | GCTGTCGTGTATTATATGTGTACCACTCTCATT |
| 1A1F99Mbtm | F99M | AATGAGAGTGGTACACATATAATACACGACAGC |
| 1A1F99Ltop | F99L | GCTGTCGTGTATTATTTATGTACCACTCTCATT |
| 1A1F99Lbtm | F99L | AATGAGAGTGGTACATAAATAATACACGACAGC |
| 1A1T402Itop | T402I | ATCAGTATCACGGCCATATCTGCCAGCATCGGA |
| 1A1T402Ibtm | T402I | TCCGATGCTGGCAGATATGGCCGTGATACTGAT |
| 1A1T402Ltop | T402L | ATCAGTATCACGGCCTTATCTGCCAGCATCGGA |
| 1A1T402Lbtm | T402L | TCCGATGCTGGCAGATAAGGCCGTGATACTGAT |
| Cloning (sgRNA) sequences | | |
| Name | Target | Sequence |
| CT110 (sgCon) | Non-targeting | GGAGGCTAAGCGTCGCAA |
| 1A1sg5 | *SLC1A1* | TCACCTGATGAGATCTAACA |

(Wu et al, 2019). Decreasing **3e**'s overall lipophilicity by the addition of polar groups reduced its potency, likely due to compromised membrane penetration and inability to access the binding site. **3e** forms two hydrogen bonds with SLC1A1: one between the core N1 and the hydroxyl group of T402; and another —mediated by a well-resolved structural water—between the aniline NH and the hydroxyl of T102 (Figs. 2F and 3A). Consistent with this, removal of the N1 hydrogen bond acceptor substantially reduced **3e**'s inhibitory potency in the previous SAR (Wu et al, 2019). The 8-bromo group of **3e** projects into a narrow hydrophobic shelf formed by I399 and A371 in HP2, and V106 in TM3. These key interactions explain why the removal of C8-substituents disrupted binding. Finally, only the C7-position of the imidazopyridine core is partially exposed to solvent, which likely explains why bulky substitutions at other positions, particularly at C5 and C6, result in reduced activity due to steric clashes with the protein wall.

Guided by these structural insights, we identified four key groups, $R_1$-$R_4$, in **3e** that were amenable to modifications (Appendix Fig. S5A) and designed a series of new BIA analogs. First, given potential concerns over the oxidative metabolism of the 2-furanyl group, we designed modifications that replaced this moiety ($R_1$). Second, we tested the possibility of replacing the aniline and phenyl substitutions at C3 ($R_2$). We maintained a hydrophobic moiety at the C8-position ($R_3$), preserving high cLogP, and retained the hydrogen bond with T402. We tested the possibility of introducing additional hydrophobic groups into the C7-position ($R_4$). Importantly, the buried C5- and C6-positions were left unsubstituted to avoid steric clashes. We computationally docked a virtual library of **3e** analog molecules into our cryo-EM structure and prioritized candidates with favorable poses and predicted free energies of binding lower than that of **3e**. Ultimately, we synthesized and evaluated ~40 BIA analogs (Appendix Tables S2 and S3).

We screened the BIA compounds using two orthogonal approaches. First, we employed the nanoDSF assay to measure the ligand-induced thermal stabilization of SLC1A1g under Na$^+$-free conditions. As expected, **3e** stabilized the carrier by ~3.5 °C; however, vehicle control (DMSO), buffer control (N-methyl-D-glucamine, NMDG), and TFB-TBOA, whose binding depends on Na$^+$, had no measurable effect (Fig. 4A). The **3e** analogs exhibited a range of stabilizing effects, with 21 candidates increasing the melting temperature by at least 1 °C (Fig. 4A; Appendix Fig. S5B). Second, we assessed the cytotoxicity of the compounds in the SLC1A1-dependent A498 cells using a high-throughput 384-well format assay over 7 days. Compared to the earlier 12-well, 14–20-day assay (Fig. 3D), the GI$_{50}$ values were generally higher in the 384-well format, likely reflecting differences in assay duration and the metabolic state of the cells. Using **3e** as a reference, we identified ~20 analogs with greater potency (Fig. 4B). From these, we selected the most potent compounds (**12, 10, 30, 38**, and **04**; Fig. 4C,D) that also stabilized SLC1A1g by more than 1 °C in the nanoDSF assays for further validation. As a negative control, we also included the highly cytotoxic compound **14**, which did not bind to SLC1A1g in the thermal shift assays (Fig. 4A), suggesting it would display off-target (i.e., non-SLC1A1-mediated) cytotoxicity.

## Identifying improved SLC1A1-selective BIA analogs

We first tested the selected analogs in two additional human RCC lines, UMRC-2 and 786O, which are less sensitive to SLC1A1 inhibition than A498 cells. Two compounds—**30** (featuring the C2-azetidine and C7-fluoro substitutions, hereafter referred to as PBJ1) and **38** (containing a C8-difluoromethyl substitution, hereafter referred to as PBJ2)—demonstrated increased cytotoxicity in these cells compared to **3e** (Fig. 5A–C). When docked into the cryo-EM structure, both PBJ1 and PBJ2 occupied the **3e** binding pocket in poses virtually identical to **3e**, while extending into adjacent available cavities (Fig. 5D). Expressing drug-resistant SLC1A1 F99M and T402I mutants in A498 cells conferred resistance to PBJ1 and PBJ2 (Fig. 5E), mirroring their resistance to **3e** (Fig. 3D). In contrast, other tested analogs showed no improvements in potency

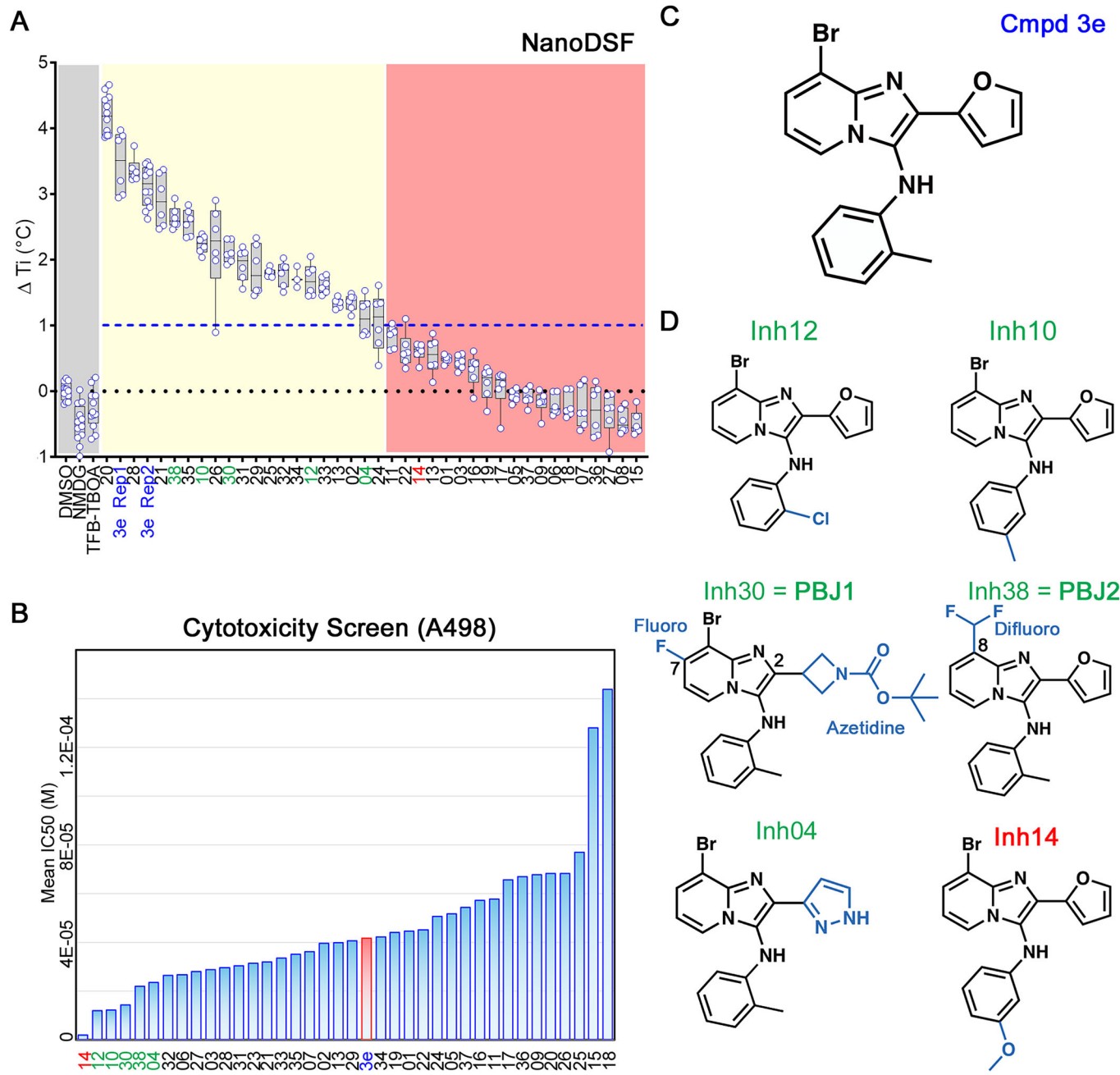

**Figure 4. Screening identifies novel BIA inhibitors.**

(A) Screening of **3e** analogs in thermal shift NanoDSF assays. Gray, yellow, and salmon regions represent, respectively, controls, compounds that stabilize SLC1A1g by more than 1 °C, suggesting binding, and compounds that do not significantly stabilize SLC1A1g, suggesting they do not bind. No data were obtained for compounds interfering with fluorescence measurements (e.g., **40** and **41**). Results are averages of at least two independent biological repeats, each comprising at least three technical repeats. The box plots representing all points from the minima to maxima, interquartile range from 25th to 75th percentile, and median central line is shown. (B) Screening of **3e** analogs in high-throughput cytotoxicity assay using kidney cancer A498 cells. Mean IC$_{50}$ values from two independent experiments are shown as blue columns and compared to **3e** control (red column). Selected compounds are colored green. (C, D) Chemical structure of the parental **3e** (C) and selected BIA compounds (D). Inhibitor **14** (red) is a highly cytotoxic off-target compound. Source data are available online for this figure.

in UMRC2 and 786O cells (Appendix Fig. S6A,B) and were cytotoxic for A498 cells expressing the drug-resistant mutants (Appendix Fig. S6C–H), suggesting that they might have off-target effects. Altogether, we discovered two new analogs— PBJ1 and PBJ2—that were more potent on-target derivatives of **3e**.

The inactivation of the von Hippel-Lindau tumor suppressor protein (pVHL), which functions as a cellular E3 ligase, and the consequent chronic activation of pVHL's cellular substrate Hypoxia Inducible Factor α-subunit (HIFα), are molecular hallmarks of ccRCC (Ricketts et al, 2018). The HIF2α isoform, in particular,

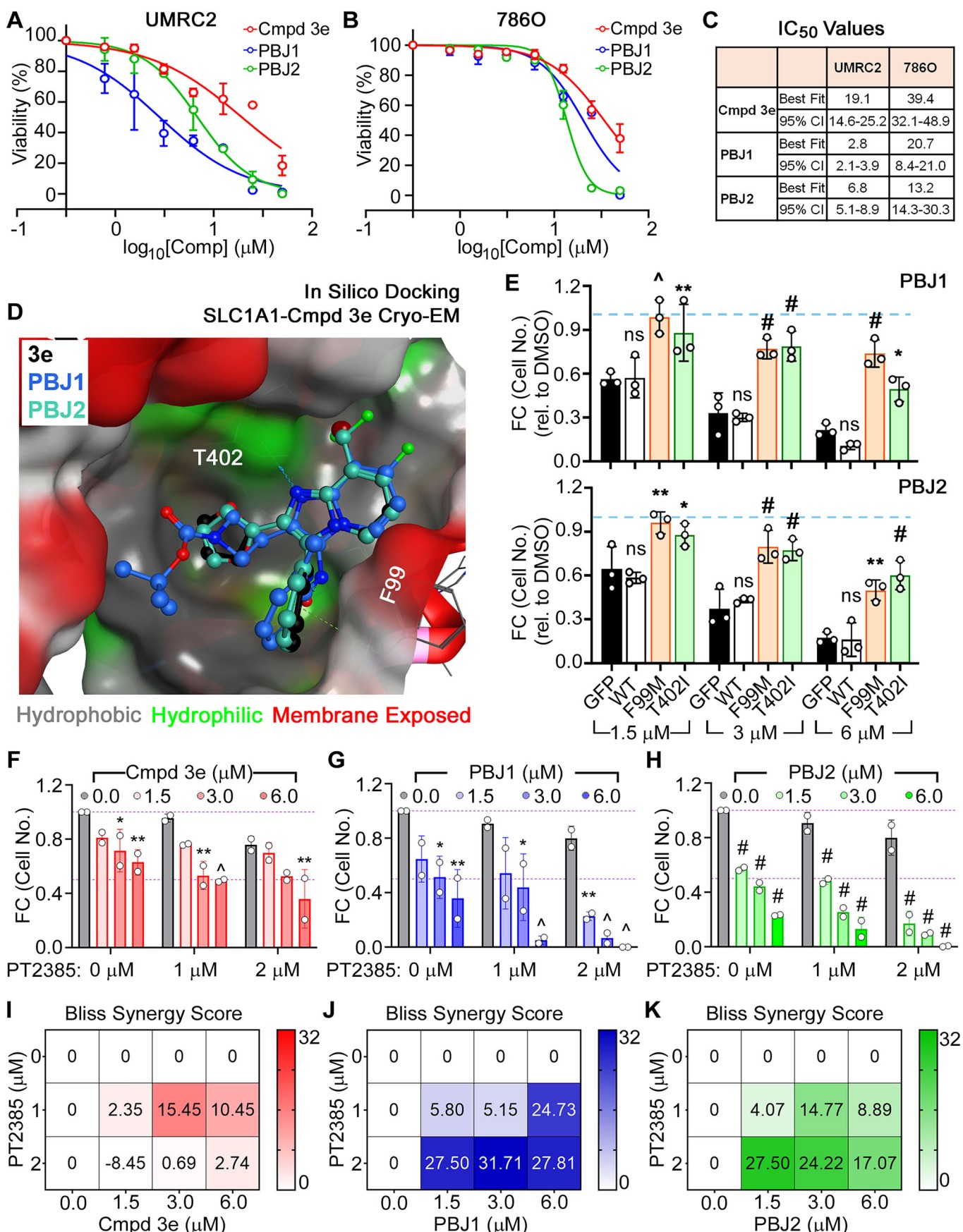

Figure 5. The antiproliferative properties of novel BIA compounds are on-target.

(A–C) Dose-dependent cytotoxic effects of the parental 3e and the novel PBJ1 and PBJ2 molecules on UMRC2 (A) and 786O (B) kidney cancer cells with fitted $IC_{50}$ values (C). (D) PBJ1 and PBJ2 docked into the 3e binding site. (E) Fold change in cell number, relative to vehicle DMSO treatment (blue dashed line), in A498 cells that were lentivirally transduced to express the indicated WT or drug-resistant mutants of SLC1A1 mutants, or GFP control, and then cultured in the presence of the indicated concentrations of PBJ1 or PBJ2 for 14–20 days. Data normalized to the DMSO arm was plotted as mean ± SD, $n = 3$, and statistically compared to the GFP arm by two-way ANOVA adjusted using Tukey's multiple comparison test. P values for PBJ1, at 1.5 µM—WT: 0.99 ns, FM: 0.0002*, TI: 0.0044**; at 3.0 µM—WT: 0.98 ns, FM: 0.0001#, TI: <0.0001#; at 6.0 µM—WT: 0.56 ns, FM: <0.0001#, TI: 0.0137*. P values for PBJ2, at 1.5 µM – WT: 0.84 ns, FM: 0.0017**, TI: 0.0237*; at 3.0 µM—WT: 0.87 ns, FM: <0.0001#, TI: 0.0001#; at 6.0 µM—WT: 0.99 ns, FM: 0.0013**, TI: <0.0001#. ns = not significant. (F–H) Fold change (FC) in cell number relative to vehicle DMSO treatment (purple dashed line at 1.0) in A498 cells that were treated with the indicated combination doses of the HIF2 inhibitor, PT2385, and the SLC1A1 inhibitors, compound 3e (F), PBJ1 (G), or PBJ2 (H). Data normalized to the DMSO arm (grey bars), $n = 2$, mean ± SD, was statistically compared using two-way ANOVA with Dunnett's multiple comparison test. P values in (F) at 0 µM PT2385—1.5: 0.16 ns, 3.0: 0.027*, 6.0: 0.0051**; 1 µM PT2385 – 1.5: 0.14 ns, 3.0: 0.0019**, 6.0: 0.001*; 2 µM PT2385—1.5: 0.86 ns, 3.0: 0.0708 ns, 6.0: 0.003**; ns = not significant. P values in (G) at 0 µM PT2385—1.5: 0.0696 ns, 3.0: 0.0128*, 6.0: 0.0019**; 1 µM PT2385—1.5: 0.602 ns, 3.0: 0.0164*, 6.0: 0.0002*; 2 µM PT2385—1.5: 0.0044**, 3.0: 0.0006*, 6.0: 0.0003*; ns = not significant. P values in (H) at 0 µM PT2385— 1.5: <0.0001#, 3.0: <0.0001#, 6.0: <0.0001#; 1 µM PT2385— 1.5: <0.0001#, 3.0: <0.0001#, 6.0: < 0.0001#; 2 µM PT2385—15: <0.0001#, 3.0: <0.0001#, 6.0: <0.0001#. (I–K) Bliss synergy scores plotted using data in (F–H), for combination doses of the HIF2 inhibitor, PT2385, and the SLC1A1 inhibitors, compound 3e (I), PBJ1 (J), or PBJ2 (K). Source data are available online for this figure.

functions as a critical oncogenic driver in ccRCC (Gordan et al, 2008; Kondo et al, 2003), and pharmacological approaches to target this protein are now FDA approved for use in ccRCC patients (Chen et al, 2016; Cho et al, 2016; Jonasch et al, 2021; Wallace et al, 2016). Notably, SLC1A1 dependence in kidney cancer is HIF-independent (Grubb et al, 2025), suggesting that targeting SLC1A1 might augment the cytotoxic effects of HIF2α blockade. To test this hypothesis, we treated the HIF2α-dependent A498 cell lines with combination doses of the HIF2α inhibitor (PT2385) and the SLC1A1 inhibitors (3e, PBJ1, and PBJ2). HIF2α inhibition causes cytotoxic effects only upon chronic treatment for ~4–6 weeks in vitro (Cho et al, 2016; Shirole et al, 2025). Consistent with this, we found relatively modest cytotoxic effects with PT2385 treatment alone (grey bars, Fig. 5F–H). However, we saw significantly higher toxicity upon combining PT2385 with the SLC1A1 inhibitors, especially the more potent PBJ1 and PBJ2, in both short-term assays with higher doses (Appendix Fig. S6I) and long-term assays with lower doses of the SLC1A1 inhibitors (Fig. 5F–H). Finally, in drug-drug interaction analysis using the SynergyFinder algorithm (Ianevski et al, 2022) we observed that these combination effects were synergistic, as evidenced by high Bliss synergy scores across various drug combinations (Fig. 5I–K).

## Improved 3e derivatives block SLC1A1 transport activity

To directly assess the inhibition of SLC1A1's transporter activity, we performed two orthogonal assays. First, we measured cellular impedance using modified culture dishes with two electrodes that measure electron flow. L-Asp and L-Glu uptake causes cellular swelling to decrease current flow. This apparent increase in impedance due to cellular swelling can be used as a proxy for substrate uptake (Fig. 6A), as previously described (Sijben et al, 2022).

We generated monoclonal HEK293 cells with doxycycline (Dox) inducible expression of SLC1A1, SLC1A2, or SLC1A3 (Fig. 6B). HEK293-SLC1A1 cells demonstrated a time-dependent increase in impedance in the presence of L-Glu, reaching equilibrium at ~1 h after substrate addition (Fig. 6C; Appendix Fig. S7A,B). Notably, these changes occurred only in the presence of Dox, which aligns with previous reports of low endogenous expression of SLC1 transporters in the HEK293 cells (Sijben et al, 2022), demonstrating their suitability for these impedance-based uptake assays. Compound 3e inhibited the uptake of 1 mM L-Glu (Fig. 6C; Appendix

Fig. S7B), 2 mM L-Glu (Appendix Fig. S7C), and 2 mM L-Asp (Appendix Fig. S7D), in a dose-dependent manner (Appendix Fig. S7E). In these assays, PBJ1 and PBJ2 showed $IC_{50}$ values of ~10 µM, comparable to 3e, while retaining >100-fold selectivity against SLC1A2 and SLC1A3 (Fig. 6D; Appendix Fig. S7F). By contrast, the TFB-TBOA competitive pan-SLC1 inhibitor (Shimamoto et al, 2004), blocked uptake by all subtypes (Fig. 6D; Appendix Fig. S7F). Thus, PBJ1 and PBJ2 retain the SLC1A1 selectivity of the parent 3e compound.

Next, we orthogonally validated our impedance assay by measuring changes in membrane potential triggered by SLC1A1-dependent L-Glu transport in the presence or absence of the SLC1A1 inhibitors. SLC1A1's ion coupled transport cycle involves the co-transport of $Na^+$ ions with L-Asp/L-Glu into the cell and an expulsion of $K^+$ ions (Fig. 1A). The ion transport cycle results in net changes in membrane potential, which can be measured using electrophysiology approaches or with specialized membrane potential dyes in Fluorescent Imaging Plate Reader (FLIPR) assays (Wu et al, 2019). Using the Dox-inducible SLC1A1 system (Fig. 6B), we found that 3e, PBJ1, and PBJ2 treatment diminished L-Glu-induced membrane potential changes in a dose-dependent manner, consistent with a blockade of the SLC1A1 transport cycle (Fig. 6E,F). As with the impedance assay, however, we did not notice any significant difference in $IC_{50}$s of the three SLC1A1 inhibitors in this acute inhibitor assay, with all molecules showing half-maximal inhibition ~2–5 µM (Appendix Fig. S7G).

## Metabolic contribution of the SLC1A1 inhibitors in kidney cancer cells

The three amino acids imported by SLC1A1—L-Asp, L-Glu, and L-Cys—are precursors for critical metabolites in cancer cells. L-Asp provides the carbon skeleton for nucleotide biosynthesis; both L-Asp and L-Glu contribute towards the tricarboxylic acid (TCA) cycle; and L-Glu and L-Cys are precursors for glutathione (reduced: GSH; oxidized: GSSG) biosynthesis (Fig. 7A). To confirm the impact of the SLC1A1 inhibitors on L-Asp uptake, we first measured intracellular L-Asp uptake in the HEK293 cells with Dox-inducible SLC1A1. To this end, we cultured the cells in the presence of Dox and the SLC1A1 blockers and then measured intracellular L-Asp pools using a colorimetric assay (Fig. 7B). This assay converts L-Asp to pyruvate and then measures pyruvate abundance using a colorimetric probe. To control for basal

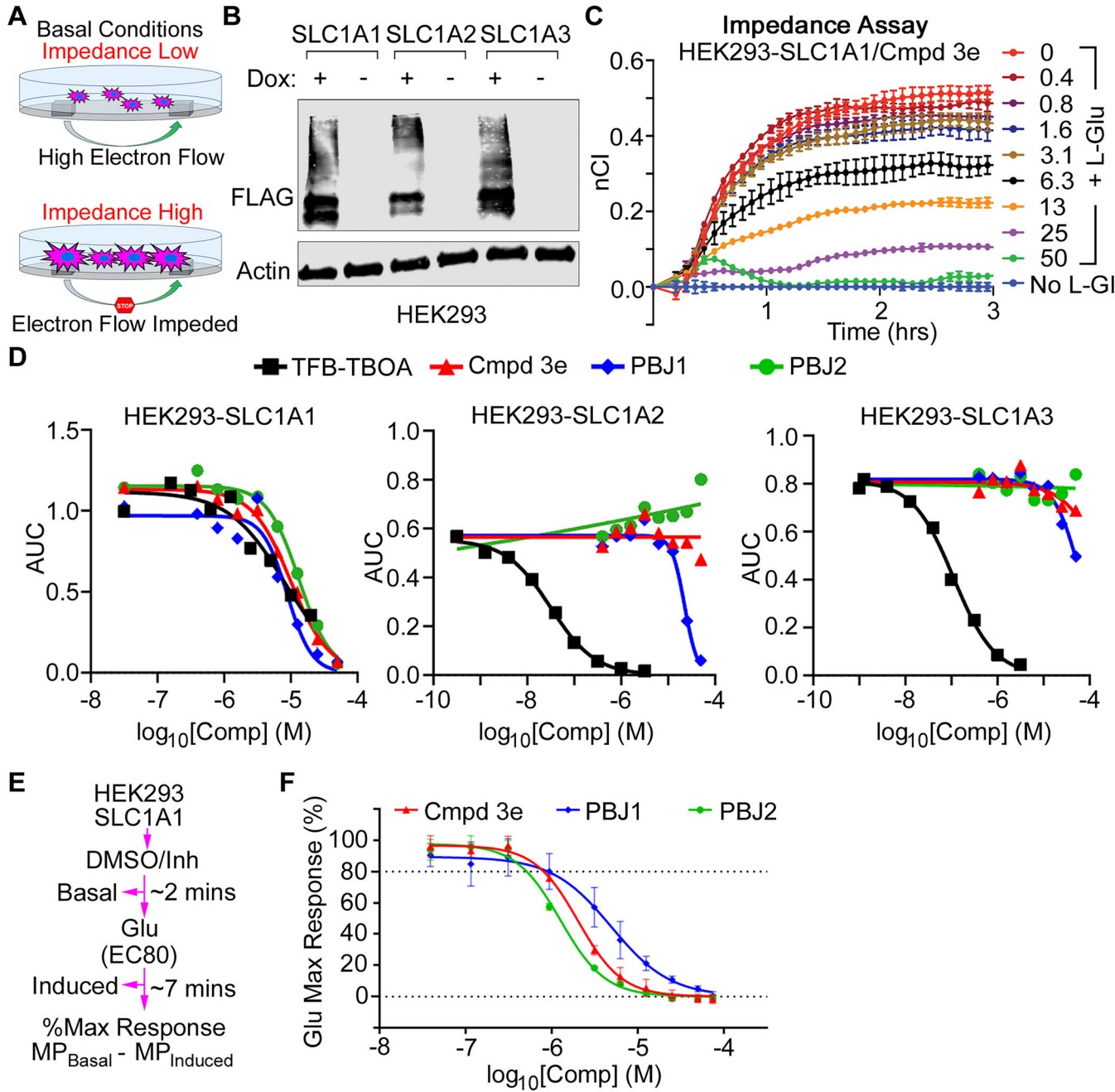

**Figure 6. Novel antiproliferative BIA compounds inhibit transport by SLC1A1.**

(A) Experimental design of the impedance assay. (B) Western blot showing SLC1 expression in HEK293 cells stably transfected with FLAG-tagged Dox-inducible versions of human *SLC1A1*, *SLC1A2*, and *SLC1A3* genes. (C) Time-dependent impedance increases in HEK293 cells expressing SLC1A1 in the presence of 1 mM ʟ-Glu and increasing concentrations of **3e**. Data represents mean ± SD from two independent replicates. (D) Dose-dependent inhibition of impedance as measured by changes in integrated Area Under the Curve (AUC) upon treatment of cells in (B, C) with the indicated BIA compounds or **3e** and TFB-TBOA as controls. (E, F) Schema of the FLIPR based membrane potential assay measuring SLC1A1 activity (E) and changes in membrane potential triggered by SLC1A1-dependent Glu uptake measured using FLIPR in the presence of the indicated concentrations of **3e**, PBJ1, and PBJ2, represented as mean ± SD, n = 12 (F). Source data are available online for this figure.

intracellular ʟ-Asp/pyruvate, we measured (and subtracted) the signal obtained from cells that were not exposed to Dox. Consistently, we observed that the SLC1A1 inhibitors reduced intracellular ʟ-Asp pools, with PBJ1 and PBJ2 having more pronounced effects (Fig. 7C).

In the A498 cells treated the SLC1A1 inhibitors we observed reduced intracellular GSH and GSSG levels (Fig. 7D,E), consistent with diminished L-Glu and L-Cys uptake, and in line with our previous metabolomics data showing a prominent reduction in TCA cycle metabolites and GSH/GSSG following **3e** treatment in

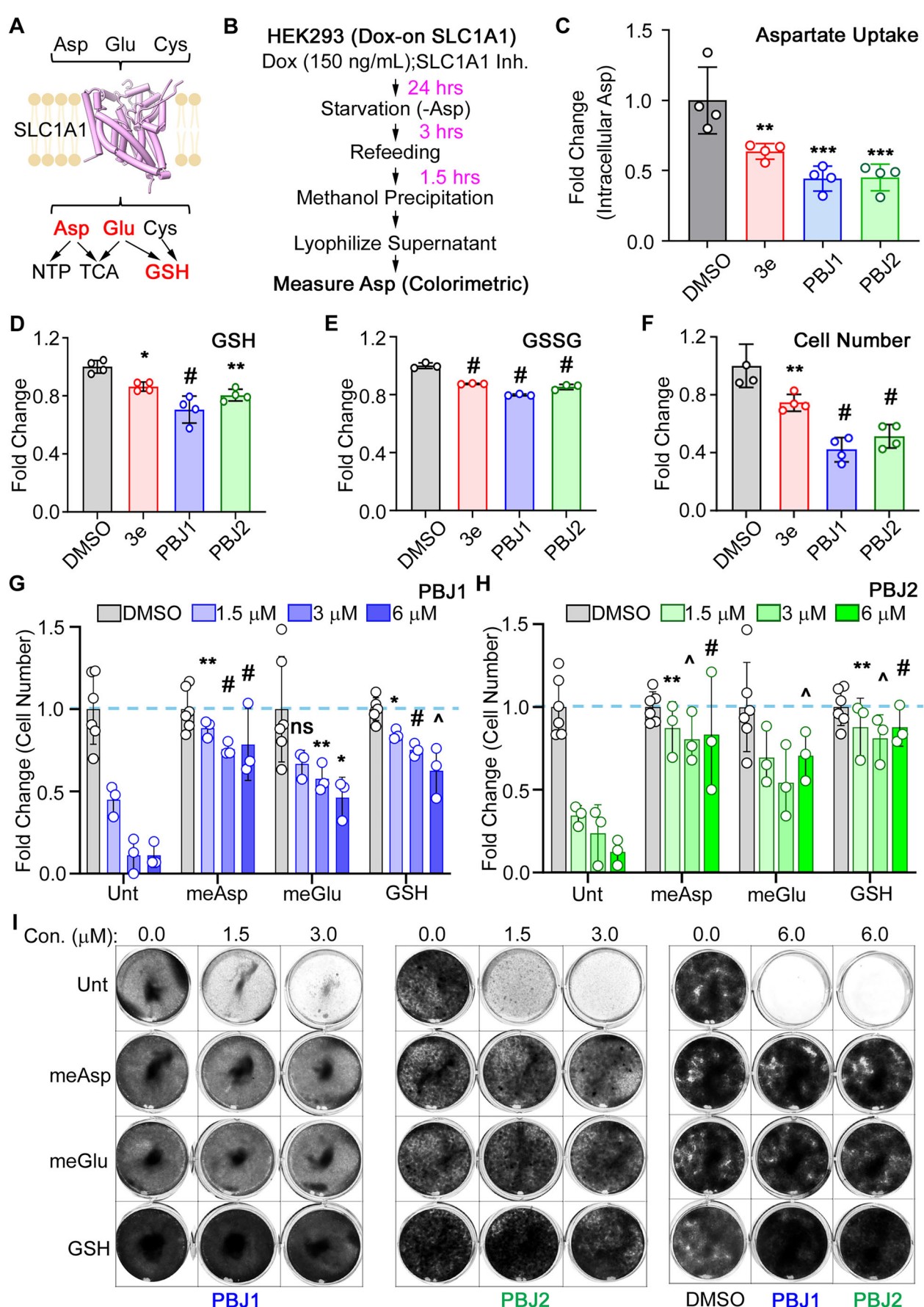

**Figure 7.  BIA compounds cause metabolic vulnerabilities in cancer cells.**

(A) SLC1A1 uptakes Asp, Glu, and Cys amino acids, which are necessary for the synthesis of nucleotides (NTPs), enter the TCA cycle, and are precursors of glutathione (GSH) biosynthesis. (B, C) Schema for measurement of SLC1A1-dependent uptake of Asp (B) and intracellular Asp levels (mean $\pm$ SD, $n = 4$) measured using a colorimetric assay kit (C) in HEK293 cells expressing Dox-inducible SLC1A1 (as described in Fig. 6B). Data compared to the DMSO arm using one-way ANOVA with Dunnett's multiple comparison test. P values: 3e: 0.0078**, PBJ1: 0.0003***, PBJ2: 0.0003***. (D–F) Intracellular levels of reduced glutathione (GSH) (D) or oxidized glutathione (GSSG) (E), and changes in cell number (F), in A498 cells that were treated with 5 $\mu$M of the indicated compounds, or DMSO control. Data plotted as mean $\pm$ SD, $n = 3$, was statistically compared to the DMSO arm by two-way ANOVA adjusted using Dunnett's multiple comparison test. P values in (D): 3e: 0.0147*, PBJ1: <0.0001#, PBJ2: 0·0011**. P values in (E): 3e: <0.0001#, PBJ1: <0.0001#, PBJ2· < 0.0001#. P values in (F): 3e: 0.0091**, PBJ1: <0.0001#, PBJ2: <0.0001#. (G–I) Cell counts measured using ViCell upon treatment with PBJ1 (G) or PBJ2 (H), and representative plates stained using crystal violet (I), in A498 cells that were grown in the presence of the indicated concentrations of PBJ1 or PBJ2, along with no treatment (Unt), 100 $\mu$M membrane-permeable methyl esters of L-Asp and L-Glu (meAsp, meGlu), or 1 mM GSH, as indicated, for 14–20 days. (G, H) Data plotted as mean $\pm$ SD, $n = 3$, was statistically compared to the same drug concentration in the Untreated (Unt) arm by two-way ANOVA adjusted using Dunnett's multiple comparison test. P values in (G) for the meAsp arm—1.5: 0.0048**, 3.0: <0.0001#, 6.0: 0.0001#; for the meGlu arm – 1.5: 0.2349 ns, 3.0: <0.0025**, 6.0: 0.0259*; for the GSH arm – 1.5: 0.0109*, 3.0: <0.0001#, 6.0: 0.0008·. P values in (H) for the meAsp arm—1.5: 0.0020**, 3.0: <0.0008·, 6.0: <0.0001#; for the meGlu arm – 1.5: 0.0512 ns, 3.0: 0.0992 ns, 6.0: 0.0006·; for the GSH arm—1.5: 0.0017**, 3.0: 0.0007·, 6.0: <0.0001#. Source data are available online for this figure.

kidney cancer cells (Grubb et al, 2025). Moreover, the reduction in GSH/GSSG levels correlated with the antiproliferative effects of these compounds (Fig. 7F). Finally, supplementation with cell-permeable methyl ester derivatives of L-Asp (meAsp), L-Glu (meGlu), or GSH rescued the fitness defects caused by PBJ1 and PBJ2 (Fig. 7G–I). Altogether, these findings supported the conclusion that the antiproliferative activity of the BIAs emerged from metabolic perturbations directly linked to SLC1A1 inhibition.

## Discussion

The SLC1 family includes five dicarboxylic amino acid transporters, EAATs, which differ in tissue distribution and physiological roles (Danbolt, 2001; Grewer et al, 2014; Kanai et al, 2013; Vandenberg and Ryan, 2013). All are expressed in the CNS, where SLC1A2/EAAT2 and SLC1A3/EAAT1 are primarily responsible for clearing synaptic L-Glu; their diminished activity leads to devastating neurological pathologies. SLC1A1/EAAT3's broader expression in peripheral tissues distinguishes it among EAATs. In the kidney proximal tubules, it mediates the reabsorption of L-Glu and L-Asp, becoming critical for sustaining the metabolic demands of kidney cancer cells. This metabolic reliance credentials SLC1A1 as a promising therapeutic target and underscores the need for SLC1A1-selective inhibitors.

Developing subtype-selective drugs for EAATs has been challenging because of the high conservation of their substrate-binding sites. Recent cryo-EM structures of SLC1A1-3 confirmed that these sites are nearly identical across paralogs, limiting the selectivity of orthosteric competitive inhibitors (Canul-Tec et al, 2017; Canul-Tec et al, 2022; Kato et al, 2022; Qiu and Boudker, 2023, 2025; Qiu et al, 2021; Zhang et al, 2022). Moreover, such compounds are often ineffective under physiological conditions where substrate concentrations are high. In contrast, allosteric SLC1 inhibitors, such as the SLC1A3-selective UCPH-101 (Abrahamsen et al, 2013; Erichsen et al, 2010), hold greater therapeutic promise because they target less conserved sites and often act via non-competitive mechanisms, enabling subtype selectivity and sustained efficacy.

Growing recognition of SLC1A1's role in human disease has spurred efforts to develop selective inhibitors, leading to the discovery of the BIA series, including compound 3e (Wu et al, 2019). However, the structural basis for 3e's SLC1A1-selective

inhibition remained unresolved. Our high-resolution cryo-EM structure revealed that 3e binds to SLC1A1 in its IF *apo* state, outside of the substrate-binding site. It engages a membrane-embedded allosteric pocket and, together with a cholesterol molecule, wedges between the transport and scaffold domains. This binding mode resembles that of UCPH-101, which binds to the SLC1A3 transporter in its OF conformation at the domain interface and is thought to inhibit transport by blocking the elevator movements of the transport domain (Canul-Tec et al, 2017). By analogy, 3e may similarly block elevator movements, arresting the transporter in the IF state. However, unlike UCPH-101, 3e also interferes with ion and substrate binding by engaging a residue that coordinates the Na+ ion.

Prior structural studies have established that SLC1A1 functions as a highly dynamic molecular machine, undergoing complex conformational rearrangements throughout its transport cycle (Fig. 1A) (Qiu and Boudker, 2023, 2025; Qiu et al, 2021). Importantly, several proposed states, such as K+-bound OF and IF conformations and K+/Na+-co-bound states (Kortzak et al, 2019; Wang et al, 2020; Wang et al, 2019), have not yet been structurally resolved. Our data nevertheless suggest that 3e selectively targets a single state or a small subset of conformations, underscoring the importance of considering the full conformational landscape of SLC1 and other dynamic membrane transporters in the development of next-generation inhibitors.

Our structures further revealed that the F99 residue, which is not conserved in other SLC1 transporters, is a major determinant of subtype selectivity. Mutating F99 produced functional transporters that could not bind 3e. These mutants, as well as mutants of T402, which is intimately involved in drug coordination, confer resistance to compound-induced cell death in RCC cell lines. These drug-resistant mutants are a gold-standard tool for demonstrating that 3e and the newly identified PBJ1 and PBJ2 analogs act on-target in cellular assays. Consistently, metabolic rescue experiments revealed that PBJ1 and PBJ2 prevent cancer cell growth by directly blocking SLC1A1-dependent metabolic programs.

PBJ1 and PBJ2 compounds emerged from our structure-guided SAR campaign. Interestingly, while they showed improved on-target cytotoxicity compared to 3e in cell assays, they did not exhibit increased affinity for SLC1A1 in nanoDSF or higher inhibitory potency in the impedance-based or FLIPR-based transport assays. This might be because the antiproliferative assays measure the impact of sustained SLC1A1 blockade over a period of

2–3 weeks. In contrast, the other assays measure more acute effects occurring on time scales of minutes to hours. Moreover, unlike the other approaches, antiproliferative assays measure the cumulative impact of blocking the uptake of all SLC1A1 substrates. Altogether, the acute binding and inhibition assays can identify bona fide SLC1A1 inhibitors but might not perfectly capture the analog's cellular potency.

PBJ1 and PBJ2 have improved certain pharmacological features over the parental **3e** molecule. For instance, PBJ1 lacks the labile furan ring, whereas PBJ2 has the difluoromethyl replacement for bromine, which improves its 'drug-like' features. Given the membrane-embedded nature of the binding pocket, further optimization of these BIA inhibitors might prove challenging. Nevertheless, our findings provide the structural basis and the rigorous validation pipeline necessary for a more comprehensive search of new chemical scaffolds that target SLC1A1.

# Methods

### Reagents and tools table

| Reagent/resource | Reference or source | Identifier or catalog number |
|---|---|---|
| **Experimental models** | | |
| FreeStyle™ 293-F Cells | Qiu and Boudker, 2023, 2025; Qiu et al, 2021 | |
| A-498 (ccRCC Cell line) | Chakraborty Lab | RRID: CVCL_1056 |
| HEK293 Cell | Chakraborty Lab | RRID: CVCL_0045 |
| HEK293T Cell | Chakraborty Lab | RRID: CVCL_0063 |
| 786-O (ccRCC Cell Line) | Chakraborty Lab | RRID: CVCL_1051 |
| UMRC-2 | Dr. Bert Zbar and Dr. Marston Linehan, NCI | RRID: CVCL_2739 |
| **Recombinant DNA** | | |
| psPAX2 | Addgene | RRID: ddgene_12260 |
| pMD2.G | Addgene | RRID: ddgene_12259 |
| pLX304 | Addgene | RRID: ddgene_25890 |
| SLC1A1 | Addgene | 161044 |
| SLC1A2 | Addgene | 161067 |
| SLC1A3 | Addgene | 161055 |
| pPB-TetOn-Puro-SLC1Ax/3xGS/3xFLAG_SLC1Ax/P2A/EGFP | Vector Builder | |
| SPB-DNA | Hera Biolabs | |
| **Antibodies** | | |
| EAAT3 (E1E6M) Rabbit Monoclonal Antibody | Cell Signaling Technology | 14501S |
| EAAT2 (E3P5K) Rabbit Monoclonal Antibody | Cell Signaling Technology | 20848S |
| EAAT1 (D44E2) Rabbit Monoclonal Antibody | Cell Signaling Technology | 5684S |
| beta-Actin (13E5) Rabbit Monoclonal Antibody | Cell Signaling Technology | 4970S |
| **Oligonucleotides and other sequence-based reagents** | | |
| Mutagenesis Primers | This study | Table 1 |
| sgRNA sequences | Grubb et al, 2025 | Table 1 |
| **Chemicals, enzymes and other reagents** | | |
| L-[$^3$H]-Asp | American Radiolabeled Chemicals, Inc | |
| SLC1A1g | Qiu and Boudker, 2023, 2025; Qiu et al, 2021 | Construct described previously |
| Lipofectamine 2000 | Invitrogen | 11668019 |
| HiFi assembly | NEB | E2621X |
| Fugene HD | Promega | E2311 |
| L-glutamate | Sigma-Aldrich | 49621 |
| Sodium Pyruvate | Gibco | 11360-070 |
| L-Glutamine | Gibco | 25030081 |
| Doxycycline | Alfa Aesar | J63805.06 |
| NaCl | Sigma | S-9888 |
| KCl | Sigma | P3911 |
| CaCl$_2$ | Sigma | C3306 |
| MgCl$_2$ | Sigma | M8266 |
| HEPES | Gibco | 15630-060 |
| D-glucose | Sigma | G7528 |
| puromycin | Phytopure™, MP Biomedicals | 0219453925 |
| CellTiter-Glo | Promega | G9241 |
| **Software** | | |
| Leginon | Suloway et al, 2005 | |
| Relion 3.1.0 | Scheres, 2012 | |
| CryoSPARC v4.2.1 | Punjani et al, 2017 | |
| https://doi.org/10.5281/zenodo.3576630 | hereafter, NUR | |
| PyEM | https://doi.org/10.5281/zenodo.3576630 | |
| CryoSPARC v4.4.1 | https://cryosparc.com/ | |
| ChimeraX | Goddard et al, 2018 | |
| COOT | Emsley et al, 2010 | |
| Phenix | Adams et al, 2010 | |
| GraphPad Prism V10 | https://www.graphpad.com/features | |
| custom analysis software | Vanderbilt Waveguide | |

| Reagent/resource | Reference or source | Identifier or catalog number |
|---|---|---|
| SynergyFinder web application | Ianevski et al, 2022 | RRID: SCR_019318 |
| Other | | |
| Tycho NT.6 | NanoTemper Technologies | |
| glow-discharged QUANTIFOIL R1.2/1.3 holey carbon-coated 300 mesh gold grids | Quantifoil, Großlöbichau | |
| scintillation counter | Beckman Coulter LS6500 | |
| Dulbecco's Modified Eagle's Medium (DMEM) | Life Technologies | 11995073 |
| 1X Penicillin-Streptomycin | Life Technologies | 15140163 |
| Fetal Bovine Serum (FBS) | Life Technologies | 10437-028 |
| Heat-inactivated Fetal Bovine Serum (FBS) | Gibco | A5256801 |
| site-directed mutagenesis Kit | Quickchange II XL; Agilent | 200521 |
| E-Plate 96 PET plate | Agilent | 300600910 |
| xCELLigence RTCA | Agilent | |
| 384 well plates | Corning PureCoat | 356719 |
| Dialyzed FBS | Gibco | 26400-044 |
| FLIPR Membrane Potential Assay Kit | Sigma | R8126 |
| Echo Qualified Low Dead Volume source plate | Labcyte | LP-0200 |
| Echo 650 | Beckman Coulter Life Sciences | |
| Multidrop Combi | ThermoFisher Scientific | |
| BioTek ELX406 TS plate washer | Agilent | |
| EnzyChrom L-Aspartate Assay Kit | BioAssay Systems | EASP-100 |
| Glutathione Colorimetric Detection Kit | Thermo Fisher | EIAGSHC |
| Microplate reader | SpectraMax i3 | |

## Protein expression and purification

The de-glycosylation (N178T, N195T) mutant of human SLC1A1/EAAT3 protein, termed SLC1A1g for short, and its K269C/W441C mutant were purified as previously described (Qiu and Boudker, 2023, 2025; Qiu et al, 2021). In brief, 1 L suspension FreeStyle™ 293-F cells at a density of 2.5 million cells per mL were transiently transfected with 3 mg DNA and 9 mg PEI 25 K (*Polysciences*). After 18 h, the cells were supplemented with 2.2 mM of valproic acid. The cells were harvested after 48 h post-transfection and resuspended in lysis buffer containing: 50 mM Tris-HCl (pH 8.0), 1 mM EDTA, 1 mM L-Asp, 1 mM Tris (2-carboxyethyl) phosphine (TCEP), 1 mM Phenylmethylsulfonyl fluoride (PMSF), and 1:200 dilution of protease inhibitor cocktail (catalog no. P8340, Sigma-Aldrich). The cells were disrupted using an EmulsiFlex-C3 homogenizer

(Avestin) and the cell debris was removed by centrifugation at $10,000 \times g$ for 15 min at 4 °C. The membrane pellets were collected by ultracentrifugation at $186,000 \times g$ for 1 h at 4 °C and solubilized in lysis buffer supplemented with 200 mM NaCl, 10% Glycerol, 1% Dodecyl-β-D-maltopyranoside (DDM, Anatrace), and 0.2% Cholesteryl hemisuccinate (CHS; Sigma-Aldrich) overnight at 4 °C for SLC1A1g. The K269C/W441C mutant was solubilized in 2% DDM with 0.4% CHS for 1 h at 4 °C. The insoluble material was removed by ultracentrifugation $186,000 \times g$ for 1 h at 4 °C; the supernatant was then incubated with Strep-Tactin Sepharose resin (GE Healthcare) for 1 h at 4 °C. The resin was washed with 8 column volumes of buffer containing 50 mM Tris-HCl (pH 8.0), 200 mM NaCl, 1 mM L-Asp, 1 mM TCEP, 5% glycerol, and 0.06% glyco-diosgenin (GDN, Anatrace), and the protein was eluted with 4 column volumes of the same buffer supplemented with 2.5–5 mM D-desthiobiotin (Sigma-Aldrich). The N-terminal Strep II and GFP tags were cleaved by incubating the protein with Prescission protease overnight at 4 °C at a 40:1 w:w ratio. The SLC1A1g protein was further purified in "apo" condition by size-exclusion chromatography (SEC) in a buffer containing: 20 mM HEPES-Tris (pH 7.4), 100 mM N-methyl-D-glucamine (NMDG) chloride, 1 mM TCEP, and 0.01% GDN. The K269C/W441C mutant was purified by SEC in a buffer containing: 20 mM HEPES-Tris (pH 7.4), 200 mM NaCl, 1 mM L-Asp, and 0.01% GDN. The fractions containing the protein were combined, concentrated to ~0.5 mg/mL, and incubated with a 20-molar excess of $HgCl_2$ for 30 min at room temperature. Following incubation, the crosslinked sample (SLC1A1g-X, for short) was purified by SEC in a buffer containing: 20 mM HEPES-Tris (pH 7.4), 300 mM KCl, and 0.01% GDN.

## Thermostability assays

Purified *apo* SLC1A1g was concentrated to ~100 μM and then diluted 20-fold in a buffer containing 50 mM HEPES-Tris pH 7.4, 100 mM NMDG-Cl, and 0.01% GDN, without or with supplementation with 100 μM **3e** or other drugs dissolved in DMSO. During the screening of **3e** analogs, the final mixtures contained 10% DMSO, whereas in the initial **3e** binding experiments, the DMSO concentration was 1%. NMDG-Cl was 200 mM in these experiments. The thermostability assay was performed using Tycho NT.6 (NanoTemper Technologies). In brief, protein samples were heated from 35 °C to 95 °C at 30 °C/min, and the intrinsic fluorescence of the protein was recorded at 330 nm and 350 nm. The ratio of fluorescent signals at 350 and 330 nm, its first derivative, and the inflection temperature (*Ti*) were calculated by the Tycho NT.6 software. All measurements were repeated on at least two independent protein preps and three technical replicates.

## Cryo-EM sample preparation and data acquisition

Purified *apo* SLC1A1g was supplemented with 200 mM NaCl and 100 μM compound **3e**, and SLC1A1g-X was supplemented with 100 μM **3e** only. Samples were concentrated to about 4 mg/mL at room temperature. 3.5 μl sample was applied to glow-discharged QUANTIFOIL R1.2/1.3 holey carbon-coated 300 mesh gold grids (Quantifoil, Großlöbichau). The grids were blotted for 3 s at room temperature and 100% humidity and plunge-frozen into liquid ethane using a Mark IV Vitrobot (*FEI*). The cryo-EM dataset for SLC1A1g with **3e** was auto-collected using Leginon (Suloway et al,

2005) on Titan Krios equipped with a 20 eV energy filter and Gatan K3 camera at the Columbia University Cryo-Electron Microscopy Center. A total of 9970 movies were collected at a nominal magnification of 100,500-fold with a calibrated pixel size of 0.83 Å in counting mode. A nominal defocus value of −1.0 to −2.0 μm and a total dose of 58 e⁻/Å² over 50 frames were applied during data collection. The data on SLC1A1g-X in KCl with **3e** were collected similarly using Titan Krios equipped with a 20 eV energy filter and Gatan K3 camera at the New York Structural Biology Center. A total of 9146 movies were collected at a nominal magnification of 81,000-fold with a calibrated pixel size of 0.43 Å in super-resolution mode. A nominal defocus value of −0.8 to −2.5 μm and a total dose of 53.9 e⁻/Å² over 55 frames were applied to the data collection.

## Cryo-EM image processing

The cryo-EM dataset for SLC1A1g with NaCl **3e** was processed using a combination of Relion 3.1.0 (Scheres, 2012) and CryoSPARC v4.2.1 (Punjani et al, 2017). The movies were aligned using Motioncor2 in Relion, and the micrographs were imported into CryoSparc. The CTF parameters of the micrographs were estimated using Path CTF, and micrographs with the CTF fit resolution worse than 6 Å were removed from the dataset. About 6.7 million particles were picked from 9594 images, and ~2.7 million particles were extracted using a box size of 160 pixels with 2× binning after particle inspection. The particles were subjected to 2D classification, and 945 K particles exhibiting secondary structure features were selected and subjected to two rounds of ab initio reconstructions. In total, 303 K particles producing a volume with the most features were further refined to 3.4 Å by nonuniform refinement (hereafter, NUR) with C1 symmetry. Separately, 162 K particles extracted from 500 micrographs were subjected to 1–10 iterations of ab initio reconstructions to generate 5 decoy volumes. After removing non-protein junk particles in the 2D selection step, about 2.5 million particles were subjected to heterogeneous refinement using one good volume (3.4 Å from the last step NUR) and 5 decoy volumes for cleaning (heterogeneous refinement cleaning, HRC). After that, 589 K selected particles were re-extracted with a box size of 320 pixels without binning and subjected to HRC with C1 and NUR with C3 symmetry. Then, the particles were imported into Relion using PyEM (https://doi.org/10.5281/zenodo.3576630) for polishing. The polished particles were reimported into CryoSparc and subjected to HRC and NUR. After 2 rounds of polishing, HRC, and NUR steps, 556 K particles were refined to 2.46 Å using C3 symmetry. Symmetry expansion and local classification were performed in CryoSPARC, with a mask applied over the protomer to resolve conformational heterogeneity. Three protomer conformations were further locally refined to 2.56 Å (*apo* IF with bound **3e** and GDN), 2.76 Å (*apo* IF with bound **3e** and cholesterol), and 2.83 Å (Na⁺-bound IF) in CryoSPARC. The cryo-EM dataset for SLC1A1g-X with KCl and **3e** was processed using CryoSPARC v4.4.1. Briefly, the movies were imported into CryoSPARC and aligned using Patch Motion Correction with 2x binning. CTF parameters were estimated using Path CTF, and micrographs with the CTF fit resolution worse than 5 Å were discarded. About 2.2 million particles were extracted from 8795 images using a box size of 150 pixels with 2× binning and were subjected to 2D classification. 912 K particles exhibiting secondary structure features were selected and subjected to one

round of ab initio reconstructions. From these, 618 K particles producing a volume with the most features were further refined to 3.64 Å by NUR with C1 symmetry (good volume). Separately, 98 K particles showing non-protein features were subjected to 1–10 iterations of ab initio reconstructions to generate 5 decoy volumes. Then, the 912 K particles were subjected to two rounds of HRC using one good volume and 5 decoy volumes. After that, 387 K selected particles were refined by NUR and re-extracted with a box size of 300 pixels without binning. The unbinned particles were subjected to HRC with C1 and NUR with C3 symmetry; the resulting 329 K particles were subjected to Reference-Based Motion Correction for polishing and finally refined to 3.11 Å by NUR with C3 symmetry. Then, symmetry expansion and local classification were performed in CryoSPARC, with a mask applied over the protomer to resolve conformational heterogeneity. Two protomer conformations were further locally refined to 3.11 Å (~80% particles, iOFS bound to K⁺) and 3.23 Å (~20% particles, *apo* OFS) in CryoSPARC.

## Model building and refinement

The IF *apo* and Na⁺-bound SLC1A1g structures (PDB codes: 6X3F and 6X2L, respectively) were fitted into EM density maps using ChimeraX (Goddard et al, 2018). The models were manually adjusted in COOT (Emsley et al, 2010) and subjected to real-space refinement in Phenix (Adams et al, 2010). Structural model validation was performed in Phenix, and structural figures were prepared using ChimeraX. The Cryo-EM maps and atomic coordinates have been deposited in the Electron Microscopy Data Bank (EMDB) and Protein Data Bank (PDB) under accession codes: SLC1A1g class A (*apo* IF with bound **3e** and cholesterol, EMD-71288, PDB-9P4X), class B (*apo* IF with bound **3e** and GDN, EMD-71289, PDB-9P4Y), and class C, (Na⁺-bound IF, EMD-71290, PDB-9P4Z), SLC1A1g-X in iOFS with bound K+ ion (EMD-75048) and apo OFS (EMD-75049).

## Radiolabeled ʟ-Asp uptake

In total, 100 mL of FreeStyle™ 293-F cells were transfected at a cell density of $1 \times 10^6$ cells/mL with a plasmid encoding SLC1A1g as described above for large-scale protein production. Forty-eight hours after transfection, cells were harvested by centrifuging at $200 \times g$ for 15 min at 4 °C and washed twice with 2 mL of resting buffer containing 11 mM HEPES-Tris (pH 7.4), 140 mM Choline chloride, 4.7 mM KCl, 2.5 mM CaCl₂, 1.2 mM MgCl₂, and 10 mM ᴅ-glucose. In total, 200 μL of cells were then diluted fivefold to a final concentration of 1.0E07 cells/mL in resting buffer and prewarmed in a water bath to 37 °C for at least 10 min. To start the reaction, 40 μL of resuspended cells were diluted fivefold in the same buffer, with 140 mM choline chloride replaced by either 20 mM NaCl and 120 mM Choline chloride or 100 mM NaCl and 40 mM Choline chloride, supplemented with appropriate concentrations of **3e**. After a 10-min pre-incubation, uptake reactions were initiated by adding 0.5 μM L-[³H]-Asp (American Radiolabeled Chemicals, Inc.) and 19.5 μM ʟ-Asp. The reactions were incubated at 37 °C for 45 s and terminated by mixing with 2 mL of ice-cold quench buffer containing 140 mM Choline chloride and 11 mM HEPES-Tris (pH 7.4). These mixtures were filtered over 0.8 μm nitrocellulose filters (MF-Millipore) and washed with 6 mL of ice-cold quench buffer. The filter was then placed into scintillation fluid, and the

retained radioactivity was quantified using a scintillation counter (Beckman Coulter LS6500). The uptake assays were performed on two independently prepared samples, each with two technical repeats.

## Cell lines

A-498 (RRID: CVCL_1056), HEK293 (RRID: CVCL_0045), HEK293T (RRID: CVCL_0063), and 786-O (RRID: CVCL_1051) cells were obtained from the American Type Culture Collection. UMRC-2 (RRID: CVCL_2739) cells were obtained from Dr. Bert Zbar and Dr. Marston Linehan (National Cancer Institute). All cells were maintained in Dulbecco's Modified Eagle's Medium (DMEM) (Life Technologies 11995073) supplemented with 10% Fetal Bovine Serum (FBS) (Life Technologies 10437-028), and 1× Penicillin–Streptomycin (Life Technologies 15140163). Cells stably expressing cDNAs or sgRNA were generated by lentiviral infection followed by selection with Puromycin (2 µg/mL) or Blasticidin (10 µg/mL), as appropriate. HEK293$^{SLC1A1/SLC1A2/SLC1A3}$ cell lines were cultured in high-glucose Dulbecco's Modified Eagle Medium (DMEM, CCF Media Core) supplemented with 10% heat-inactivated Fetal Bovine Serum (FBS) (Gibco A5256801), 5 U/mL Penicillin/Streptomycin (Sigma P3032 & Gibco 11860), 2 mM L-glutamine (CCF Media Core), and 0.8 mg/mL puromycin (Phytopure™, MP Biomedicals 0219453925). All cells were grown at 37 °C in 5% $CO_2$. Cell lines were periodically tested for mycoplasma and cell line identity was confirmed using STR analysis by the Cleveland Clinic's Cell and Media Core facility.

## Lentiviral infections

Lentiviral particles to introduce cDNAs or sgRNAs into cells were produced from HEK293T packaging cells. In a standard experiment, we seeded ~1.6E06−2.0E06 HEK293T cells into a 6-cm dish and transfected these cells with a transfection mix that contained 3 µg total DNA [1.5 µg lentiviral plasmid + 1.5 µg helper plasmids – psPAX2 (RRID: Addgene_12260) and pMD2.G (RRID: Addgene_12259) mixed in a 3:1 ratio] and 9 µl lipofectamine 2000 (Invitrogen 11668019). The transfection mixtures were prepared and incubated at RT for 20 min and then added dropwise onto cells. Cells were incubated for sixteen hours, following which the media was changed. Viral supernatant was collected at 48- and 72-h post-transfection, combined, and filtered (0.45 µm). Aliquots were stored frozen at −80 °C. To transduce recipient cells, we seeded six-well plates with 1.5E05−2.0E05 cells and transduced them with 0.5 mL of the lentiviral supernatant in the presence of 8 µg/mL polybrene. The plates were returned to the incubator for 16 h, after which the media was replaced, and cells were grown for an additional 24 h prior to selection with the appropriate antibiotic.

## Generation of SLC1A1 mutant constructs

Entry vectors and pLX304 constructs encoding sg5-resistant versions of SLC1A1 were previously described (Grubb et al, 2025). Mutations encoding F99A, F99M, F99L, T402I, and T402L were generated in the sg5-resistant backbone either using overlap-extension PCR or by site-directed mutagenesis (Quickchange II XL; Agilent 200521) using the oligos described in the primer list. All mutant clones were verified by Sanger sequencing. Engineered clones were lentivirally transduced into recipient cells using the procedures described above. For the genetic rescue experiments to assay SLC1A1 functionality, cells were first engineered to express the sgRNA-resistant versions of either WT or mutant SLC1A1 (or GFP, as a control) in the pLX304 backbone (RRID: Addgene_25890). Following selection with blasticidin (10 µg/mL, 7–10 days), cells were lentivirally transduced with CRISPR/Cas9 constructs targeting SLC1A1 (1A1sg5) or a non-targeting control and selected with puromycin (2 µg/mL, 3 days). The sgRNA sequences are provided in the primer list. Following the second selection, equal number of viable cells were seeded at low density (typically, 100–200 cells/well of a 12-well plate) and cultured for 1–2 weeks post selection before staining the wells with crystal violet (0.25% in 20% methanol, stained overnight at 4 °C).

## High-throughput screening assay for compound 3e and its analogs

A twofold serial dilution of each test compound was prepared in 100% DMSO starting at 25 mM. White tissue culture 384-well plates were first pre-filled with 30 µL of complete DMEM with 1 mM L-Glu. Next, 100 nL of each compound dilution was acoustically dispensed into the plates in triplicate with a Labcyte Echo 550; vehicle (0.2% DMSO final) and compound **3e** served as negative and positive controls, respectively. Finally, A 498 were seeded by adding 20 µL of a $2.5 \times 10^4$ cells /mL suspension (≈500 cells per well) using an E1 ClipTip electronic multichannel pipette, bringing the total well volume to 50 µL while maintaining 0.2% DMSO. Plates were incubated for seven days at 37 °C, 5% CO2. Cell viability was measured by adding 25 µL CellTiter Glo® 2.0 reagent (Promega G7571) to each well, shaking for 5 min at 600 rpm, incubating 10 min at room temperature, and recording luminescence on a Revvity EnVision multimode plate reader. Luminescence in the vehicle-treated wells was normalized to 100% and used to perform regression analysis and establish each inhibitor's GI50, using the built-in algorithms in Graphpad Prism (e.g., log-inhibitor versus response – variable slope).

## Cytotoxicity assays using SLC1A1 inhibitors

Cytotoxicity assays were used to evaluate the potency and on-target effects of the new SLC1A1 inhibitors (Cmpd **3e** analogs). To compare potency in human clear cell Renal Cell Carcinoma (ccRCC) cell lines, such as A 498, UMRC2, and 786 O, cells were seeded into 96-well plates (200 cells/well in 100 µL DMEM) and allowed to attach overnight. Serial twofold dilutions of each test inhibitor were prepared in 100% DMSO and diluted into complete medium to yield final inhibitor concentrations between 0.8 and 50 µM, each in 0.1% (v/v) DMSO. Overnight culture media was aspirated and 200 µL of the appropriate drug containing media was gently added to each well.

To test on-target effects of each inhibitor, A-498 cells expressing either wild-type, F99M, or T402I SLC1A1 mutants (or GFP expressing controls) were seeded (100 cells/well in 1 mL volume) in a 12-well tissue culture plate. After 16 h to allow attachment, cells were exposed either to test compounds (final concentrations as indicated) or to 0.1% (v/v) DMSO vehicle (negative control). The medium—including fresh compound or DMSO—was replaced every three days to ensure continuous treatment. After 21 days, colonies were trypsinised and viable cell numbers were counted using a Vi-Cell to assess the effects of treatment. Three independent biological replicates were statistically compared.

For metabolic rescue experiments, 3.0E05 A-498 cells were cultured in the presence of esterified L-Asp (meAsp, 100 μM), esterified L-Glu (meGlu, 100 μM), reduced Glutathione (GSH, 1 mM), or untreated (controls) for 3 days. The cells were then trypsinized, counted, and reseeded (1000 cells/well in a 12-well plate). The cells were cultured in the presence of the indicated supplements (or untreated controls) and PBJ1 or PBJ2 (0, 1.5, 3.0, or 6.0 μM for 2–3 weeks with media changes every 3–4 days. Fresh inhibitor and metabolite supplements were added at each media change. Cell counts were measured using the Vi-Cell counter, and three independent biological replicates were statistically compared to evaluate differences.

## Generation of HEK293<sup>SLC1A1/SLC1A2/SLC1A3</sup> monoclonal cell lines

Full-length, human, mammalian codon-optimized SLC1A1 (#161044), SLC1A2 (#161067), and SLC1A3 (#161055) entry clones were obtained from AddGene. The SLC1Ax coding sequences were PCR amplified with compatible overhangs and cloned by a HiFi assembly (NEB) into a custom pPB-TetOn-Puro-SLC1Ax/3xGS/3xFLAG_SLC1Ax/P2A/EGFP (Vector Builder). This construct allows stable integration, via the PiggyBac® system, of a Dox-inducible SLC1Ax-3xFLAG cassette alongside a post-translationally released (P2A) EGFP. All three constructs were confirmed by whole-plasmid sequencing (Plasmidsaurus).

Parental HEK293 cells were co-transfected with pPB-TetOn-Puro-SLC1Ax/3xGS/3xFLAG_SLC1Ax/P2A/EGFP plasmids and SPB-DNA (Hera Biolabs) at a 3:1 ratio with Fugene HD (Promega) per manufacturer's instructions. After 24 h, the transfection media was replaced with full cell culture media supplemented with 0.8 mg/mL puromycin, and stably transfected pools were established after 5 days of selection.

For the establishment of monoclonal lines, each SLC1Ax selected pool was first enriched for high doxycycline-induced GFP expression by flow sorting using a ThermoFisher Bigfoot Spectral Cell Sorter. Each pool was seeded in full cell culture media in 2 T-175 flasks. At 80% confluence, cells were treated with 0.15 mg/mL doxycycline or vehicle for 24 h. In total, $3 \times 10^7$ cells were harvested from each flask and resuspended to $1 \times 10^7$ cells/mL in cell sorting buffer: 1× Phosphate buffered saline ($Ca^{+2}/Mg^{+2}$ free), 1 mM EDTA, 25 mM HEPES pH 7.0, 1% FBS. A separate T-75 flask of parental HEK293 cells was also harvested and resuspended in cell sorting buffer to facilitate the definition of the (+) GFP gate. The top 4–6% GFP-expressing cells within the (+) GFP gate were sorted and plated at 1 cell/well in full cell culture media supplemented with 0.8 mg/mL puromycin in 96-well plates. In total, 10–15 clones each for SLC1A1, SLC1A2, and SLC1A3 were expanded to T-75 flasks and tested for responsiveness in cell impedance measurements upon addition of 1 mM L-glutamate. SLC1A1 Monoclonal #6, SLC1A2 Monoclonal #5, and SLC1A3 Monoclonal #1 were used in reported impedance assays.

## Impedance assay

The impedance assay was conducted as previously described (Sijben et al, 2022). Briefly, HEK293<sup>SLC1A1</sup>, HEK293<sup>SLC1A2</sup>, and HEK293<sup>SLC1A3</sup> monoclonal lines were passaged in complete DMEM media (2 mM L-glutamine). On Day 1 of the assay, a modified assay media

containing DMEM with only 1 mM (0.5×) L-glutamine was prepared, and 50 μL was added to each well of an E-Plate 96 PET plate (Agilent, 300600910) and incubated at RT for at least 30 min. HEK293<sup>SLC1Ax</sup> cells were split as standard into assay media and further diluted to a final concentration of 1.2E06 cells/mL. Once the cells were prepared, the E-Plate containing only media was placed in an xCELLigence RTCA (Agilent), and a background reading was taken. The plate was removed from the block, and 50 μL of cell suspension was added to each well, at a final concentration of 6.0E04 cells/well in 100 μL of assay media. Cells were incubated at room temperature for 30 min and then placed back onto the xCELLigence overnight. Readings were taken every 5 min until the completion of the assay.

On the morning of Day 2, a 450 ng/mL solution of doxycycline was prepared by adding 0.675 μL of a 10 mg/mL DMSO stock of doxycycline to 15 mL of DMEM complete media containing 10% FBS, 50 I.U. pen/strep, but without L-glutamine. 50 μL of doxycycline was added per well to achieve a final concentration of 150 ng/mL, and SLC1Ax induction was allowed to proceed for 6–8 h. Compound serial dilutions (1:2 in 100% DMSO) were prepared at a 1000× top dose. In all, 0.8 μL of the serially diluted compounds were added to 200 μL of L-glutamine-free assay media and mixed well. In total, 50 μL of the compounds diluted in media (now 4×) were then added to the plate in technical duplicates. To columns 1 and 12, media containing the equivalent amount of DMSO was added, and the plate was returned to the xCELLigence overnight.

After 16–20 h (Day 3), 10 μL of a 21 mM (21×) PBS stock of L-glutamate (Sigma-Aldrich 49621) was prepared fresh and added to a final concentration of 1 mM to columns 2 through 12. In total, 10 μL of PBS only was added to column 1 as a vehicle control. For the comparison of L-Glu vs. L-Asp, a 42 mM stock was prepared, and the final concentration of agonist was 2 mM. The E-Plate was returned to the xCELLigence, and readings were continued for three hours post-stimulation.

For data analysis, raw cell index (CI) values were normalized to time = 0 (the last reading before L-Glu addition) within each well, and then the background vehicle response in PBS-stimulated cells (column 1) was subtracted to generate nCI values. The AUC from the nCI time-course (over 3 h) was extracted and plotted against log[compound] to extract $IC_{50}$ values by non-linear regression using GraphPad Prism (v10). The maximum response in L-Glu stimulated cells (column 12) was set as the top of the curve.

## Panoptic FLIPR membrane potential assay

The HEK293-SLC1A1 cells were cultured in growth media containing DMEM (Gibco #11960-044), 10% heat-Inactivated FBS (Gibco #A5256801), 50 unit/mL Pen and 50 μg/mL Strep (Gibco #15140-122), 1 mM Sodium Pyruvate (Gibco #11360-070), and 1 mM L-Glutamine (Gibco #25030081). Additionally, 0.83 μg/mL of Puromycin (MP Biomedicals #0219453925) was added freshly to the growth media.

Cells were seeded into amine-coated 384-well plates (Corning PureCoat #356719) at 15,000 cells/20 μL/well, with the plating media. The plating media contains the same components as the growth media except that 10% heat-Inactivated FBS was replaced by 10% Dialyzed FBS (Gibco #26400-044). Where relevant, Doxycycline (Alfa Aesar #J63805.06) was added to the cells at a

final concentration of 0.15 µg/mL, following which cells were incubated at 37 °C, 5% $CO_2$ overnight.

Test compounds, L-Glu, and the membrane potential dye powder (Molecular Devices FLIPR Membrane Potential Assay Kit, RED, Explorer format # R8126) were prepared in KREBS buffer [140 mM NaCl (Sigma #S-9888), 4.7 mM KCl (Sigma #P3911), 2.5 mM $CaCl_2$ (Sigma #C3306), 1.2 mM of $MgCl_2$ (Sigma #M8266), 11 mM of HEPES (Gibco #15630-060), and 10 mM D-glucose (Sigma #G7528)].

The SLC1A1 inhibitors (10 mM in DMSO) were loaded into an Echo Qualified Low Dead Volume source plate (Labcyte #LP-0200) using an acoustic liquid handler Echo 650 (Beckman Coulter Life Sciences, Indianapolis, IN, 46268) in a 10-point dose range (38 nM to 75 µM) in quadruplet. The membrane potential dye was diluted by twofold using Krebs buffer, resulting in a (0.5×) dye concentration and 40 µL/well of this compound dilution buffer was dispensed into the Echo-completed destination compound plate using the Multidrop Combi (ThermoFisher Scientific, Waltham, MA). The compound plate was shaken at 500 rpm for 5 min on a Teleshaker and then centrifuged at 1000 rpm for 1 min. 35 µL/well of the Glu $EC_0$ (Krebs buffer only), 5× $EC_{80}$ (300 µM Glu), or 5× $E_{MAX}$ (1 mM Glu) was loaded into the stimulus plate based on the plate map, using a multichannel pipette.

The cell plate was washed on the BioTek ELX406 TS plate washer (Agilent) in three cycles with 80 µL/well/cycle of Krebs buffer, leaving residual volume of 20 µL/well. In all, 20 µL/well of 1× dye loading solution was dispensed into the cell plate and incubated at room temperature for 45 min. After dye incubation, the buffer in the cell plate was aspirated down to 20 µL/well by ELX406 TS.

Panoptic tip box, cell plate, destination compound plate, and the glutamate stimulus plate were put into designated nests on Panoptic. The assay protocol was set with excitation at 517/20 nm, emission at 580/60 nm, gain at 1, and exposure 150 ms. After 124 s of baseline reading, 20 µL/well of 2× inhibitor was injected and read for 133 s (giving the basal membrane potential up to 257 s from the beginning of the assay) before the second injection of 10 µL/well of 5× glutamate from the stimulus plate and read continuously for an additional 607 s (giving the induced membrane potential). Fluorescent values were reduced using custom analysis software (Vanderbilt Waveguide) in a series of operations as follows: Fluorescent ratio calculated as values of the measurements over time divided by the initial fluorescent value at time 0. The maximum value during course of the $EC_{80}$ (300 µM) glutamate stimulation (257–700 s) minus the minimum value 5 s before glutamate is added (250–255 s). This reduced value is normalized to the experimental plate mean response to 1 mM glutamate representing the $E_{MAX}$. The percent maximum response of the concentration series of each test compound was fit using a 4PL model (GraphPad Prism) to estimate the potency reported as $IC_{50}$.

## Intracellular L-aspartate (L-Asp) uptake assay

HEK293 cells expressing the doxycycline-inducible SLC1A1 construct were seeded in 6-cm dishes (3.0E06 cells per dish). SLC1A1 expression was induced with doxycycline (150 ng/mL), with parallel non-induced controls. Concurrently, cells were treated overnight with Cmpd **3e**, PBJ1 or PBJ2 (10 µM). On the assay day, cells were washed once with PBS and incubated in glutamine-free DMEM supplemented with 2% dialyzed FBS and 1× Penicillin–Streptomycin for 3 h at 37 °C. Cells were subsequently pulsed with L-Asp (100 µM) for 1.5 h at 37 °C, maintaining the SLC1A1 inhibitors (10 µM) during starvation and adding aspartate.

Following the L-Asp pulse, cells were rinsed with normal saline (0.9% NaCl), and plates were immediately placed on dry ice. Samples were deproteinized by adding 1 mL ice-cold 80% (v/v) methanol (−80 °C). Plates were incubated at −80 °C for 20 min, scraped on dry ice, and lysates were transferred to microcentrifuge tubes. Samples were centrifuged at 14,000 ×g for 5 min at 4 °C to remove debris. A total of 950 µL of the supernatant was transferred to fresh tubes and dried using a SpeedVac concentrator. Dried extracts were resuspended in 105 µL PBS.

Intracellular L-Asp was quantified using the EnzyChrom L-Aspartate Assay Kit (EASP-100; BioAssay Systems) according to the manufacturer's instructions. Briefly, 25 µL of each sample was loaded in duplicate into a 96-well plate together with L-Asp standards (0–400 µM). Sample blanks were prepared using working reagent lacking AST enzyme. Working reagent was prepared by combining Developer, AST enzyme, ODC enzyme, co-substrate, and dye reagent, and 75 µL was added per well. Plates were incubated for 60 min at 25 °C protected from light, and absorbance was measured at 570 nm. L-Asp concentrations were determined from a standard curve after blank subtraction. The L-Asp levels in cells that were not treated by Dox were subtracted to calculate the amount of L-Asp in the presence or absence of the three inhibitors.

## Drug combination assay and synergy analysis

For short-term (7 day) assays, A498 cells (1000 cells/well in a 96-well plate) were exposed to the HIF2α inhibitor PT2385 (0, 1, and 2 µM) in the presence of 10 µM of Cmpd **3e**, PBJ1, or PBJ2. At the end point, viable cell numbers were measured using CellTiter-Glo (Promega G9241). For the long-term (2–3 week) assays, A498 cells (200 cells per well in 12-well plates) were exposed to Cmpd **3e**, PBJ1, or PBJ2 (0, 1.5, 3, and 6 µM), each tested in combination with the HIF2α inhibitor PT2385 (0, 1, and 2 µM). Culture medium with fresh inhibitors was replenished every 3 days. At the end point, viable cell numbers were quantified using a Vi-Cell automated cell counter (Beckman Coulter). Cell viability was expressed relative to DMSO-treated controls. Experiments were performed as three independent biological replicates. Drug interaction was evaluated using the SynergyFinder web application (RRID: SCR_019318) (Ianevski et al, 2022) with the Bliss independence model and default settings.

## Glutathione detection

Glutathione measurements were done as per the manufacturer's instructions (Thermo Fisher Glutathione Colorimetric Detection Kit, Cat. # EIAGSHC). Briefly, A-498 cells (1.5E05 in a 10 cm dish) were treated with either Cmpd **3e**, PBJ1, or PBJ2 [each at 5 µM in 0.1% (v/v) DMSO (vehicle control)] for 5 days. Cells were washed once with 1× PBS, trypsinized, and then harvested (300× g, 5 min, 4 °C). Equal number of viable cells (5.0E05 per condition) were transferred to pre-chilled tubes, washed once in ice-cold 1× PBS and immediately deproteinised in ice-cold 5% sulfosalicylic acid (SSA). Cell lysates, generated by vortexing and sonication (3" pulse

at 30% amplitude), were incubated on ice for 10 min and then clarified (14,000×*g*, 10 min, 4 °C). The supernatant was split into two pre-chilled tubes to measure total (GSH) and oxidized (GSSG) glutathione levels.

GSH samples were processed directly, whereas GSSG samples were treated with 2-vinylpyridine (2VP; 5 μL 2VP solution per 250 μL lysate) for 1 h at room temperature to derivatize free thiols and leave only oxidized glutathione. Both samples were first diluted 1:5 with Assay Buffer (final 1% SSA) and then another time at 1:4 with Sample Diluent buffer (final 1:20 dilution). In total, 50 μL diluted samples were combined with 25 μL Colorimetric Detection Reagent and 25 μL Reaction Mixture (NADPH + glutathione reductase). Plates were mixed gently and incubated 20 min at room temperature, then read at 405 nm on a SpectraMax i3 microplate reader. In parallel, a standard curve was generated for GSH (0–25 μM) and GSSG (0–12.5 μM) and used to calculate absolute concentrations of GSH and GSSG in the cell lysates.

## Statistical analysis

The sample sizes were determined based on prior experience working with these cell culture models. All experiments were typically repeated three (or more) times. One exception was the cytotoxicity screen, which was repeated twice because of the scale and cost of this experiment. The users were not blinded to the treatment groups. The statistical tests and sample sizes used in each experiment are outlined in the figure legends. Unless otherwise specified, the samples sizes indicate biological replicates. All tests were corrected for multiple testing, as necessary. Statistical significance was calculated using the Graphpad Prism software with appropriate corrections for multiple testing (as specified in the figure legends). Exact *P* values were not reported below 0.0001 in this software, and these are marked as <0.0001# in our data.

## Procedures for the synthesis of Cmpd 3e and analogs

### General scheme

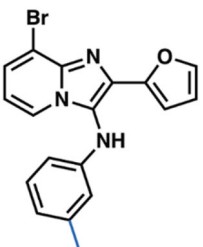

Synthesis of SLC1A1 (EAAT3) imidazopyridine cmpd **3e** and analogs reported herein were prepared according to the previously reported microwave irradiation protocol (Wu et al, 2019). Procedures, characterization data, and purity are provided for compounds **4, 10, 12, 14, 30** (PBJ1), **38** (PBJ2). Additional analogs outside these compounds were prepared following a similar condensation microwave heating protocol, beginning with commercial or known isocyanides 1, aldehydes 2, and aminopyridines

3. The final imidazopyridines **1-38** had analytical purity >95% as judged by LC-MS and 1H NMR.

### 8-Bromo-2-(1H-pyrazol-3-yl)-N-(o-tolyl)imidazo[1,2-a]pyridin-3-amine (4)

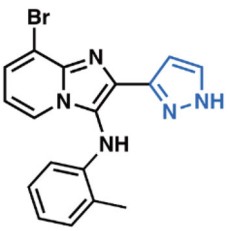

In a microwave reaction vial 1-isocyano-2-methylbenzene (102 mg, 0.87 mmol, 1.0 eq), 1*H*-pyrazole-3-carbaldehyde (84 mg, 0.87 mmol, 1.0 eq), and 3-bromopyridin-2-amine (150 mg, 0.87 mmol, 1.0 eq) were combined. The mixture was stirred neat at 160 °C for 1 h under microwave radiation. The crude product was purified by silica gel chromatography to give 8-bromo-2-(1H-pyrazol-3-yl)-*N*-(*o*-tolyl)imidazo[1,2-*a*]pyridin-3-amine (51.3 mg, 16%) as yellow solid: LCMS (ESI): *m/z* = 368.0, [M + H]⁺, *R*ₜ = 1.495 min; >95% (210, 254 nm); ¹H NMR (400 MHz, DMSO) δ 12.5 (m, 1H), 8.19 (s, 1H), 7.72 (d, J = 6.8 Hz, 1H), 7.50 (s, 1H), 7.14 (d, J = 6.8 Hz, 1H), 6.87 (dt, J = 14.5, 7.0 Hz, 2H), 6.69 (t, J = 7.4 Hz, 1H), 5.90 (d, J = 8.0 Hz, 1H), 2.40 (s, 3H).

### 8-Bromo-2-(furan-2-yl)-N-(m-tolyl)imidazo[1,2-a]pyridin-3-amine (10)

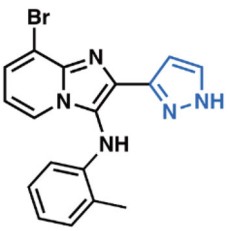

Step 1. Preparation of N-m-tolylformamide.. A mixture of 3-methoxyaniline (2 g, 18.7 mmol, 1 eq), formic acid (30 mL), and sodium formate (12.7 g, 186.6 mmol, 10 eq) in a 100 mL round-bottom flask was stirred with a magnetic stir bar at room temperature. The progress of the reaction was monitored by TLC. When the reaction reached completion, 30 mL of EtOAc was added into the system, and the organic solvent was then washed with H₂O (3 × 50 mL) and a saturated solution of NaHCO₃(50 mL). After drying with Na₂SO₄, the solvent was removed under reduced pressure. The resulting residue (890 mg, 35%) containing the *N-m*-tolylformamide was taken on to the next step without additional purification: LCMS (ESI): *m/z* = 152.0, [M + H]⁺, *R*ₜ = 0.635 min; >90% (210, 254 nm).

Step 2. Preparation of 1-isocyano-3-methylbenzene.. To a solution of *N-m*-tolylformamide (800 mg, 5.92 mmol, 1 eq) and Et₃N (1.98 g, 19.53 mmol, 3.3 eq) in anhydrous DCM (10 mL) was added dropwise POCl₃ (998.2 mg, 6.51 mmol, 1.1 eq) at 0 °C under a nitrogen atmosphere over 35 min. Then the reaction mixture was stirred at 0 °C for 1 h. After completion, the reaction mixture was poured into a saturated solution of NaHCO₃ at 0 °C. The mixture was extracted by DCM. The organic phase was dried over Na₂SO₄ and the solvent was removed under reduced pressure. The crude product was purified by silica gel column chromatography to give the title product as a yellow green liquid (600 mg, 86%) which was used directly in step 3 without further purification.

Step 3. Procedure for preparation of 8-bromo-2-(furan-2-yl)-N-(m-tolyl)imidazo[1,2-a]pyridin-3-amine.. To 1-isocyano-3-methylbenzene (203.1 mg, 1.73 mmol, 1.0 eq) from step 2, furan-2-carbaldehyde (166.6 mg,1.73 mmol, 1.0 eq), and 3-bromopyridin-2-amine (300 mg, 1.73 mmol, 1.0 eq) were mixed together in a microwave reaction vial. The mixture was stirred neat at 160 °C for 1 h under microwave radiation. The crude product was purified by silica gel chromatography to give 8-bromo-2-(furan-2-yl)-N-(*m*-tolyl)imidazo[1,2-*a*]pyridin-3-amine (53.3 mg, 8%) as white solid: LCMS (ESI): *m/z* = 369.9, [M + 3H]⁺, $R_t$ = 1.810 min; >90% (210, 254 nm); ¹H NMR (400 MHz, DMSO) δ 8.27 (s, 1H), 8.01 (d, J = 5.9 Hz, 1H), 7.76 (d, J = 1.0 Hz, 1H), 7.68 (d, J = 6.5 Hz, 1H), 7.03 (s, 1H), 6.88 (s, 1H), 6.71 (s, 1H), 6.58 (dd, J = 3.4, 1.8 Hz, 1H), 6.33 (dd, J = 8.0, 2.0 Hz, 1H), 6.09 (d, J = 2.1 Hz, 1H), 6.05 (d, J = 7.9 Hz, 1H), 3.63 (s, 3H).

### 8-Bromo-N-(2-chlorophenyl)-2-(furan-2-yl)imidazo[1,2-a]pyridin-3-amine (12)

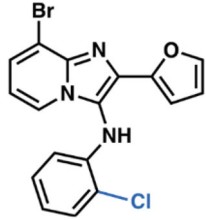

Step 1. Preparation of N-(2-chlorophenyl)formamide.. A mixture of 2-chloroaniline (3 g, 23.6 mmol, 1 eq), formic acid (30 mL), and sodium formate (1.6 g, 23 mmol, 1 eq) in a 50 mL round-bottom flask was stirred with a magnetic stir bar at room temperature. The progress of the reaction was monitored by TLC. When the reaction finished, 30 mL of EtOAc was added into the system, and the organic solvent was then washed with H₂O (3 × 50 mL) and a saturated solution of NaHCO₃ (50 mL). After drying with Na₂SO₄, the solvent was removed under reduced pressure. The resulting residue (2 g, 56%) containing the *N*-(2-chlorophenyl)formamide was taken on to the next step without further purification: LCMS (ESI): *m/z* = 156.1, [M + H]⁺, $R_t$ = 1.330 min; >95% (210, 254 nm).

Step 2. Preparation of 1-chloro-2-isocyanobenzene.. To a solution of N-(2-chlorophenyl)formamide (3.5 g, 22.5 mmol, 1 eq) and Et₃N (7.5 g, 74.5 mmol, 3.3 eq) in anhydrous DCM (40 mL) was added dropwise POCl₃ (3.8 g, 24.8 mmol, 1.1 eq) at 0 °C under a nitrogen atmosphere over 35 min. Then the reaction mixture was stirred at 0 °C for 1 h. After completion, the reaction mixture was poured onto a saturated solution of NaHCO₃ at 0 °C. The mixture was extracted by DCM. The organic phase was dried over Na₂SO₄ and the solvent was removed under reduced pressure. The crude product 1-chloro-2-isocyanobenzene was purified by silica gel column chromatography to give the product as a yellow green liquid (2.4 g, 80%). The isocyanide was used directly in step 3 without further purification.

Step 3. Preparation of 8-bromo-N-(2-chlorophenyl)-2-(furan-2-yl) imidazo[1,2-a]pyridin-3-amine.. To 1-chloro-2-isocyanobenzene (500 mg, 3.64 mmol, 1.0 eq) from step 2, furan-2-carbaldehyde (350 mg, 3.64 mmol, 1.0 eq), and 3-bromopyridin-2-amine (627 mg, 3.64 mmol, 1.0 eq) were mixed together in a microwave reaction vial. The mixture was heated neat at 160 °C for 1 h under microwave radiation. The crude product was purified by silica gel chromatography to give 8-bromo-*N*-(2-chlorophenyl)-2-(furan-2-yl)imidazo[1,2-*a*]pyridin-3-amine (45 mg, <5%) as white solid: LCMS (ESI): *m/z* = 389.8, [M + H]⁺, $R_t$ = 1.855 min; >95% (210, 254 nm); ¹H NMR (400 MHz, DMSO) δ 8.06 (d, J = 5.9 Hz, 1H), 7.98 (s, 1H), 7.74 (d, J = 1.1 Hz, 1H), 7.70 (d, J = 6.6 Hz, 1H), 7.42 (dd, J = 7.9, 1.4 Hz, 1H), 6.98 (s, 1H), 6.88 (t, J = 7.0 Hz, 1H), 6.77 (s, 1H), 6.63 (d, J = 3.3 Hz, 1H), 6.56 (dd, J = 3.3, 1.8 Hz, 1H), 6.11 (dd, J = 8.2, 1.4 Hz, 1H).

### 8-Bromo-2-(furan-2-yl)-N-(3-methoxyphenyl)imidazo[1,2-a]pyridin-3-amine (14)

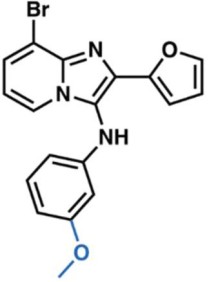

Step 1. Preparation of N-(3-methoxyphenyl)formamide.. A mixture of 3-methoxyaniline (3 g, 24.39 mmol, 1 eq), formic acid (30 mL), and sodium formate (16 g, 243.9 mmol, 10 eq) in a 50 mL round-bottom flask was stirred with a magnet at room temperature. The progress of the reaction was monitored by TLC. When the reaction finished, 30 mL of EtOAc was added into the system, and the organic solvent was then washed with H₂O (3 × 50 mL) and a saturated solution of NaHCO₃ (50 mL). After drying with Na₂SO₄, the solvent was removed under reduced pressure. The resulting residue (2.4 g, 15.89 mmol, 66.6%) containing the *N*-(3-methoxyphenyl)formamide was taken on

to the next step without additional purification: LCMS (ESI): $m/z = 152.0$, $[M + H]^+$, $R_t = 0.635$ min; >95% (210, 254 nm).

Step 2. Preparation of starting material 1-isocyano-3-methoxybenzene.. To a solution of *N*-(3-methoxyphenyl)formamide (2.4 g, 15.8 mmol, 1 eq) and Et$_3$N (5.29 g, 52.4 mmol, 3.3 eq) in anhydrous DCM (40 mL) was added dropwise POCl$_3$ (2.67 g, 17.4 mmol, 1.1 eq) at 0 °C under a nitrogen atmosphere over 35 min. Then the reaction mixture was stirred at 0 °C for 1 h. After completion, the reaction mixture was poured into a saturated solution of NaHCO$_3$ at 0 °C. Then the mixture was extracted by DCM. The organic phase was dried over Na$_2$SO$_{4,}$ and the solvent was removed under reduced pressure. The crude product was purified by silica gel column chromatography to give the product as a yellow-green liquid (2.0 g, 95%). The isocyanide was used directly in step 3 without further purification.

Step 3. Preparation of 8-bromo-2-(furan-2-yl)-N-(3-methoxyphenyl) imidazo[1,2-a]pyridin-3-amine.. To a solution of 1-isocyano-3-methoxybenzene (200 mg, 1.5 mmol, 1.0 eq) from step 2 above, furan-2-carbaldehyde (144 mg,1.5 mmol, 1.0 eq), and 3-bromopyridin-2-amine (258 mg, 1.5 mmol, 1.0 eq) were combined in a microwave reaction vial. The mixture was heated neat at 160 °C for 1 h under microwave radiation with stirring. The crude product was purified by silica gel chromatography to give title compound 8-bromo-2-(furan-2-yl)-*N*-(3-methoxyphenyl)imidazo[1,2-a]pyridin-3-amine (40 mg, 7%) as a white solid: LCMS (ESI): $m/z = 385.9$, $[M + H + 2]^+$, $R_t = 1.557$ min; >95% (210, 254 nm); $^1$H NMR (400 MHz, DMSO) δ 8.27 (s, 1H), 8.01 (d, J = 5.9 Hz, 1H), 7.76 (d, J = 1.0 Hz, 1H), 7.68 (d, J = 6.5 Hz, 1H), 7.03 (s, 1H), 6.88 (s, 1H), 6.71 (s, 1H), 6.58 (dd, J = 3.4, 1.8 Hz, 1H), 6.33 (dd, J = 8.0, 2.0 Hz, 1H), 6.09 (d, J = 2.1 Hz, 1H), 6.05 (d, J = 7.9 Hz, 1H), 3.63 (s, 3H).

### Tert-butyl 3-(8-bromo-7-fluoro-3-(o-tolylamino)imidazo[1,2-a] pyridin-2-yl)azetidine-1-carboxylate (PBJ1)

A mixture of 1-isocyano-2-methylbenzene (122.7 mg, 1.59 mmol, 1.05 eq), *tert*-butyl-3-formylazetidine-1-carboxylate (194 mg, 1.05 mmol, 1.0 eq), and 3-bromo-4-fluoropyridin-2-amine (200 mg, 1.05 mmol, 1.0 eq) was combined in a microwave reaction vial. The mixture was stirred at 160 °C for 1 h under microwave radiation. The crude product was purified by RP-HPLC to give the title compound (102.2 mg, 6%) as yellow solid: LCMS (ESI): $m/z = 477.0$, $[M + 3H]^+$, $R_t = 1.557$ min; >95% (210, 254 nm); $^1$H NMR (400 MHz, DMSO) δ 12.5 (m, 1H), 8.19 (s, 1H), 7.72 (d, J = 6.8 Hz, 1H), 7.50 (s, 1H), 7.14 (d, J = 6.8 Hz, 1H), 6.87 (dt, J = 14.5, 7.0 Hz, 2H), 6.69 (t, J = 7.4 Hz, 1H), 5.90 (d, J = 8.0 Hz, 1H), 2.40 (s, 3H).

### 8-(Difluoromethyl)-2-(furan-2-yl)-N-(o-tolyl)imidazo[1,2-a]pyridin-3-amine (PBJ2)

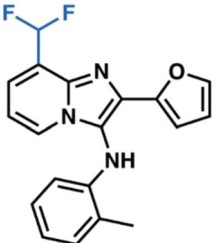

A mixture of 1-isocyano-2-methylbenzene (150 mg, 1.28 mmol), furan-2-carbaldehyde (123 mg, 1.28 mmol), and 3-(difluoromethyl)pyridin-2-amine (185 mg, 1.28 mmol) were combined in a microwave reaction vial and heated neat at 150 °C for 1 h under microwave radiation. After completion, the reaction was cooled and purified by flash chromatography (EtOAc/hexanes). The crude product containing fractions were concentrated and further purified by HPLC. The final product was isolated as a yellow solid to give 8-(difluoromethyl)-2-(furan-2-yl)-*N*-(o-tolyl)imidazo[1,2-a]pyridin-3-amine (34.79 mg, 8%): LCMS (ESI): $m/z = 340.1$, $[M + H]^+$, $R_t = 1.35$ min; >95% (210, 254 nm); $^1$H NMR (400 MHz, DMSO) δ 8.14 (d, J = 6.6 Hz, 1H), 7.74 (s, 1H), 7.63 (s, 0.24H), 7.59 (d, J = 6.7 Hz, 1H), 7.49 (s, 1.5H), 7.36 (s, 0.26H), 7.15 (d, J = 7.3 Hz, 1H), 7.03 (t, J = 6.9 Hz, 1H), 6.84 (t, J = 7.6 Hz, 1H), 6.67 (t, J = 7.3 Hz, 1H), 6.62 (d, J = 3.2 Hz, 1H), 6.55 (dd, J = 3.3, 1.8 Hz, 1H), 5.87 (d, J = 7.9 Hz, 1H), 2.41 (s, 3H).

## Data availability

All the data necessary to evaluate the conclusions of the manuscript are provided in the paper and/or in the Appendix. The Cryo-EM maps and atomic coordinates have been deposited in the Electron Microscopy Data Bank (EMDB) and Protein Data Bank (PDB) under accession codes: EMD-71288 (SLC1A1g protomer with **3e** and cholesterol, class A), corresponding to PDB-9P4X; EMD-71289 (SLC1A1g protomer with **3e** and GDN, class B), corresponding to PDB-9P4Y; EMD-71290 (Na$^+$-bound SLC1A1g protomer), corresponding to PDB-9P4Z; EMD-75048 (SLC1A1 in the presence of 300 mM KCl and **3e**); and EMD-75049 (SLC1A1 in the OF state in the presence of 300 mM KCl and **3e**).

The source data of this paper are collected in the following database record: biostudies:S-SCDT-10_1038-S44318-026-00776-2.

## Peer review information

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

## Acknowledgements

We thank Carl Fluck at the Weill Cornell Medicine Electron Microscopy facility for the initial grid screening and preliminary data collection, and Robert Grassucci, Zhening Zhang, and Yen-hong Kao at Columbia University Cryo-Electron Microscopy Center for high-resolution data collection. NanoDSF experiments were performed on the Tycho NT.6 instrument in the Molecular Biophysics core facility at Weill Cornell Medicine under the leadership of Dr. Radda Rusinova. This work was supported by the following grants to AAC.: seed money from the Cleveland Clinic Foundation, the DoD KCRP-ECI award (W81XWH-20-1-0804), and the Velosano pilot award. Additionally, AAC, OB, and SRS were supported with a DoD-KCRP-IDA (HT9425-25-1-0761). PK was supported by a postdoctoral fellowship from Velosano. JAC, NSW, TR, CMG and SRS were supported by funds from Shared Laboratory Services at the Lerner Research Institute. OB was supported by HHMI and the National Institute of Neurological Disorders and Stroke (R37NS085318).

## Author contributions

**Pooneh Koochaki**: Conceptualization; Data curation; Formal analysis; Investigation; Writing—original draft; Writing—review and editing. **Biao Qiu**: Conceptualization; Data curation; Formal analysis; Investigation; Writing—original draft; Writing—review and editing. **Jesse A Coker**: Conceptualization; Data curation; Formal analysis; Investigation; Writing—

review and editing. **Alexander Earsley**: Conceptualization; Data curation; Formal analysis. **Nancy S Wang**: Conceptualization; Data curation; Formal analysis. **Todd Romigh**: Conceptualization; Data curation; Formal analysis. **Christopher M Goins**: Conceptualization; Data curation. **Carleigh Salem**: Data curation; Formal analysis. **Dehui Mi**: Data curation; Formal analysis. **Emily Days**: Data curation; Formal analysis. **Joshua A Bauer**: Data curation; Formal analysis; Supervision. **Shaun R Stauffer**: Conceptualization; Data curation; Supervision; Funding acquisition; Writing—original draft; Writing—review and editing. **Olga Boudker**: Conceptualization; Data curation; Supervision; Funding acquisition; Writing—original draft; Writing—review and editing. **Abhishek A Chakraborty**: Conceptualization; Data curation; Supervision; Funding acquisition; Writing—original draft; Writing—review and editing.

Source data underlying figure panels in this paper may have individual authorship assigned. Where available, figure panel/source data authorship is listed in the following database record: biostudies:S-SCDT-10_1038-S44318-026-00776-2.

## Disclosure and competing interests statement

The authors declare no competing interests.

