## [Peer Review File · The EMBO Journal]

Structure-Guided Optimization of SLC1A1/EAAT3-Selective Inhibitors Targeting Renal Cancer Metabolism

Pooneh Koochaki, Biao Qiu, Jesse Coker, Alexander Earsley, Nancy Wang, Todd Romigh, Christopher Goins, Carleigh Salem, Dehui Mi, Emily Days, Joshua Bauer, Shaun Stauffer, Olga Boudker, and Abhishek Chakraborty

Corresponding author(s): Abhishek Chakraborty (chakraa@ccf.org) , Olga Boudker (olb2003@med.cornell.edu), Shaun Stauffer (stauffs2@ccf.org)

Review Timeline:

Submission Date:	5th Aug 25
Editorial Decision:	8th Sep 25
Revision Received:	23rd Jan 26
Editorial Decision:	24th Feb 26
Revision Received:	1st Mar 26
Accepted:	19th Mar 26

Editor: William Teale

Transaction Report:

Dear Dr. Chakraborty,

Thank you again for the submission of your manuscript entitled "Mechanism and Structure-Guided Optimization of SLC1A1/EAAT3-Selective Inhibitors in Kidney Cancer" and for your patience during the review process. We have now received reports from two referees, which I copy below.

As you can see from their comments, both thought that your manuscript could represent a timely and well-directed study. Both of them point out some issues that will require your attention before your manuscript can be published in The EMBO Journal. I am particularly concerned about Referee #1's insistence that the binding of 3e to SLC1A1/EAAT3 and its mode of inhibition be more thoroughly characterised. Addressing these points will need additional experimentation.

Based on the overall interest expressed in the reports, however, I would like to invite you to address the comments of all referees in a revised version of the manuscript. I should add that it is The EMBO Journal policy to allow only a single major round of revision and that it is therefore important to resolve the main concerns at this stage. I believe the concerns of the referees are reasonable and addressable, but please contact me if you have any questions, need further input on the referee comments or if you anticipate any problems in addressing any of their points. I am always available on Zoom should you wish to discuss the referees' reports or your responses to them; I would be particularly interested in discussing the feasibility of your addressing Reviewer #1's comments. Please, follow the instructions below when preparing your manuscript for resubmission.

I would also like to point out that as a matter of policy, competing manuscripts published during this period will not be taken into consideration in our assessment of the novelty presented by your study ("scooping" protection). We have extended this 'scooping protection policy' beyond the usual 3 month revision timeline to cover the period required for a full revision to address the essential experimental issues. Please contact me if you see a paper with related content published elsewhere to discuss the appropriate course of action.

Again, please contact me at any time during revision if you need any help or have further questions.

Thank you very much again for the opportunity to consider your work for publication. I look forward to your revision.

Best regards,

William

William Teale, Ph.D.
Editor
The EMBO Journal

When submitting your revised manuscript, please carefully review the instructions below and include the following items:

- 1) a .docx formatted version of the manuscript text (including legends for main figures, EV figures and tables). Please make sure that the changes are highlighted to be clearly visible.
- 2) individual production quality figure files as .eps, .tif, .jpg (one file per figure).
- 3) a .docx formatted letter INCLUDING the reviewers' reports and your detailed point-by-point response to their comments. As part of the EMBO Press transparent editorial process, the point-by-point response is part of the Review Process File (RPF), which will be published alongside your paper.
- 4) a complete author checklist, which you can download from our author guidelines ([https://wol-prod-cdn.literatumonline.com/pb-assets/embo-site/Author Checklist%20-%20EMBO%20J-1561436015657.xlsx](https://wol-prod-cdn.literatumonline.com/pb-assets/embo-site/Author%20Checklist%20-%20EMBO%20J-1561436015657.xlsx)). Please insert information in the checklist that is also reflected in the manuscript. The completed author checklist will also be part of the RPF.
- 5) Please note that all corresponding authors are required to supply an ORCID ID for their name upon submission of a revised manuscript.
- 6) We require a 'Data Availability' section after the Materials and Methods. Before submitting your revision, primary datasets

produced in this study need to be deposited in an appropriate public database, and the accession numbers and database listed under 'Data Availability'. Please remember to provide a reviewer password if the datasets are not yet public (see <https://www.embopress.org/page/journal/14602075/authorguide#datadeposition>). If no data deposition in external databases is needed for this paper, please then state in this section: This study includes no data deposited in external repositories. Note that the Data Availability Section is restricted to new primary data that are part of this study.

Note - All links should resolve to a page where the data can be accessed.

8) For data quantification: please specify the name of the statistical test used to generate error bars and P values, the number (n) of independent experiments (specify technical or biological replicates) underlying each data point and the test used to calculate p-values in each figure legend. The figure legends should contain a basic description of n, P and the test applied. Graphs must include a description of the bars and the error bars (s.d., s.e.m.).

9) We would also encourage you to include the source data for figure panels that show essential data. Numerical data can be provided as individual .xls or .csv files (including a tab describing the data). For 'blots' or microscopy, uncropped images should be submitted (using a zip archive or a single pdf per main figure if multiple images need to be supplied for one panel). Additional information on source data and instruction on how to label the files are available at .

10) We replaced Supplementary Information with Expanded View (EV) Figures and Tables that are collapsible/expandable online (see examples in <https://www.embopress.org/doi/10.15252/embj.201695874>). A maximum of 5 EV Figures can be typeset. EV Figures should be cited as 'Figure EV1, Figure EV2" etc. in the text and their respective legends should be included in the main text after the legends of regular figures.

12) Our journal encourages inclusion of *data citations in the reference list* to directly cite datasets that were re-used and obtained from public databases. Data citations in the article text are distinct from normal bibliographical citations and should directly link to the database records from which the data can be accessed. In the main text, data citations are formatted as follows: "Data ref: Smith et al, 2001" or "Data ref: NCBI Sequence Read Archive PRJNA342805, 2017". In the Reference list, data citations must be labeled with "[DATASET]". A data reference must provide the database name, accession number/identifiers and a resolvable link to the landing page from which the data can be accessed at the end of the reference. Further instructions are available at .

13) In order to increase the reproducibility and reach of your work, The EMBO Journal includes a table of reagents that were used in the study. Please provide this along with your revisions.

We realize that it is difficult to revise to a specific deadline. In the interest of protecting the conceptual advance provided by the work, we recommend a revision within 3 months (7th Dec 2025). Please discuss the revision progress ahead of this time with the editor if you require more time to complete the revisions. Use the link below to submit your revision:

Referee #1:

This manuscript by Koochaki reports the development of novel treatment options for Renal Cell Carcinomas (RCCs). Based on earlier work of the Chakraborty group that upregulation of SLC1A1/EAAT3 glutamate transporter is associated with this disease the authors combine cell biology, structural biology and medical chemistry to improve an existing EAAT3 blocker (named 3e) that may help future cancer treatment in certain patients.

This is a certainly an important project; the use of high profile basic research to answer pressing medical questions is very appealing, and the combination of multiple approaches is very impressive. However, I find the description of the blockade mechanism not sufficient.

Major concerns

1. Compound optimization is based on cryo-EM imaging of human EAAT3 in the presence of 200 mM NaCl and 100 μ M 3e. The authors report a density consistent with 3e only in apo protomers and conclude that 3e binds to apo inward-facing SLC1A1g. I find this a surprising conclusion, since I would expect that transporters only shortly assume an apo IF state during transport. I would guess that K⁺ binds immediately after dissociation of the last Na⁺ or - in agreement with recent electrophysiological evidence (DOI: 10.1074/jbc.RA119.009421) that there might exist K⁺/Na⁺ co-binding state.

This reasoning raises the question whether 3e might also or even predominantly bind to a K⁺- bound state, and this particular aspect is not sufficiently addressed. The authors demonstrate that occupation of one K⁺ site does not interfere with the E3 binding site in the apo state, however, this does not seem sufficient to me. I find the evidence for this particular K⁺ binding site not very strong, and other K⁺ binding sites have been postulated (DOI: 10.1085/jgp.202012577 and DOI: 10.15252/embj.2019101468); these sites and potential roles in 3e binding should be discussed here.

2. I am missing a more detailed description of the binding mechanism. The authors use only indirect approaches, such as thermal shift binding assays or impedance assay, which do not provide important information about the binding process and the transporters state of binding. For example, the authors suggest that "3e either selectively binds to apo SLC1A1g or displaces bound Na⁺"; I would insist that the authors distinguish between these two options with experimental approach. Is binding of 3e reduced at higher Na⁺ concentrations?

3. Moreover, what is the evidence that the compound binds from the cytoplasm? At present, the authors only provide a hypothetical binding process, diffusion of 3e across the membrane and subsequent binding from the insight, I would urge the

authors to support this or another binding path with experiments. Can the authors exclude that EAAT3 binds 3e from the outside and translocates to IF? One could learn a lot with cell electrophysiological experiments to investigate the 3e block.

Minor concerns

4. What is the evidence that 3e indeed blocks the transporter? How does 3e modify the EAAT3 anion channel? Can the authors provide on and off rates of binding?
5. I remember from my Renal Physiology class that EAAT3 is exclusively expressed in the proximal tubules of the nephron. I am therefore surprised that it plays such a prominent role in the most common form of adult kidney cancer, and I have to admit that I kept thinking about the general role of EAAT3 in kidney during reading the whole manuscript. Does it mean that Renal Cell Carcinomas originate from the proximal tubule? Could the authors provide some explanation why a proximal tubule transport plays a role in renal cancer and speculate whether it is also expressed in other forms of cancers?
6. Moreover, since EMBO J is not a clinical journal and the prospective audience will mostly not consist of oncologists, I would suggest to give a better introduction into the whole idea, i.e. can we expect that an EAAT3 can cure renal cancer and what about other forms of cancer (see comment 5.)?
7. I find it concerning that the scientific basis for this study, i.e. that EAAT3 plays a role in kidney cancer has not been accepted for publication, but has only posted on biorxiv.

Referee #2:

Renal Cell Carcinomas (RCCs) depend metabolically on the trimeric sodium-coupled aspartate and glutamate transporter SLC1A1/EAAT3. However, pharmacologically targeting SLC1A1 is challenging because the amino acid residues interacting with the substrates and the co transported ions in the binding pocket of EAAT3 are identical those of the related EAAT1 and EAAT2 transporters, which are essential for function of the brain. Recently a compound, termed 3e was found which selectively inhibits EAAT3. To obtain more potent inhibitors, cryogenic electron microscopy (cryo-EM) structure of human SLC1A1/EAAT3 bound to compound 3e was determined. The structure reveals the transporter in an inward-facing apo state where 3e is located between the transport and scaffold domains and identifies a molecular determinant of the selectivity of 3e for EAAT3. Structure-guided modifications of the 3e scaffold led to the identification of two SLC1A1 selective inhibitors with improved cancer cell-killing potency and suggest a potential for the development of SLC1A1-targeted therapies.

This is a well executed multi-disciplinary study, combining structural biology, synthetic organic chemistry and cell biology. Despite all the work involved, the two new inhibitors show only modest improvement of cancer cell-killing. However, the study provides some new mechanistic insights, such as the location of the inhibitor site which partly overlaps with the Na³ site at T102 as well as the identification of F99 as an important determinant of selectivity. The Authors use impedance as a read-out of transport and show that 3e and the new compounds selectively inhibit EAAT3. However, impedance measurements are not very familiar to most readers, including those in the transporter field. This technique should be clearly described, indicating its advantages and drawbacks relative to the established methodology. What the Authors show as a cartoon in Fig. 6A without any further explanation is clearly not sufficient. In addition, conventional transport measurements such as radioactive uptake should be done in parallel with some of the impedance measurements to underscore its validity as readout of transport. Moreover, it is important to determine kinetic parameters to establish the mechanism of inhibition by the compounds.

A minor point is the mentioning of F99 as "the determinant" of selectivity (line 350). To support such a statement, phenylalanine should be introduced at the corresponding positions of EAAT1 and EAAT2 to see if now these transporters become sensitive to 3e. That would be great, but without such proof one should say that F99 is "a major determinant" of selectivity. In addition, because of the proximity of the inhibitor to the Na³ site, it would be worthwhile to check if the inhibition of transport by 3e is increased at lower sodium concentrations.

Reviewer 1

This manuscript by Koochaki reports the development of novel treatment options for Renal Cell Carcinomas (RCCs). Based on earlier work of the Chakraborty group that upregulation of SLC1A1/EAAT3 glutamate transporter is associated with this disease the authors combine cell biology, structural biology and medical chemistry to improve an existing EAAT3 blocker (named 3e) that may help future cancer treatment in certain patients.

This is a certainly an important project; the use of high profile basic research to answer pressing medical questions is very appealing, and the combination of multiple approaches is very impressive. However, I find the description of the blockade mechanism not sufficient.

We thank the Reviewer for appreciating the significance of the work. The insightful comments have helped us to improve the quality of our work. We have performed several new experiments and present a new cryo-EM dataset to address the reviewer's concerns about the inhibition mechanism. In addition, although not requested by the reviewers, **we have performed cell-killing experiments showing that EAAT3 inhibitors, especially the newly developed PBJ1 and 2 inhibitors, act synergistically with HIF2 α inhibitors in killing the kidney cancer cells**, highlighting the potential physiological significance of these compounds (please see new **Figs. 5F to 5K** and new **Appendix Fig. S6I**)

Major concerns

1. Compound optimization is based on cryo-EM imaging of human EAAT3 in the presence of 200 mM NaCl and 100 μ M 3e. The authors report a density consistent with 3e only in apo protomers and conclude that 3e binds to apo inward-facing SLC1A1g. I find this a surprising conclusion, since I would expect that transporters only shortly assume an apo IF state during transport. I would guess that K⁺ binds immediately after dissociation of the last Na⁺ or - in agreement with recent electrophysiological evidence (DOI: 10.1074/jbc.RA119.009421) that there might exist K⁺/Na⁺ co-binding state.

We were also surprised by these findings. The caveat of structural studies, relevant to this comment and the following comments by the Reviewer, is that they are performed in detergent solution, where the protein's behavior might not fully recapitulate its behavior in cell membranes. Nevertheless, our cryo-EM structures are consistent with the thermal-shift binding experiments, which show that Na⁺ inhibits 3e binding. We have now performed a reverse experiment, titrating Na⁺ and showing that increased 3e concentrations inhibit Na⁺ binding (**Response Fig. 1**, new **Fig. 1G**). Furthermore, structure-based mutagenesis tested in cell-based assays and thermal shift assay, and SAR analysis support the observed binding site and the bound inhibitor pose. We discuss these results more extensively in the revised manuscript in the context of the DOI: 10.1074/jbc.RA119.009421 paper.

Response Fig. 1: 3e inhibits Na⁺ binding.

Thermal shifts induced by increasing Na⁺ concentrations, reflecting Na⁺ binding, are diminished in the presence of increasing concentrations of 3e, suggesting that 3e binds the apo state, competing with Na⁺ binding.

This reasoning raises the question whether **3e** might also or even predominantly bind to a K⁺-bound state, and this particular aspect is not sufficiently addressed. The authors demonstrate that occupation of one K⁺ site does not interfere with the **3e** binding site in the apo state, however, this does not seem sufficient to me. I find the evidence for this particular K⁺ binding site not very strong, and other K⁺ binding sites have been postulated (DOI: 10.1085/jgp.202012577 and DOI: 10.15252/embj.2019101468); these sites and potential roles in **3e** binding should be discussed here.

Structurally, we have not observed K⁺-bound IFS in K⁺ concentrations up to 300 mM (DOI: 10.1038/s41467-023-38120-5). We interpret this result as indicating that K⁺ binds weakly to the IFS (at least in detergent). Thus, we could not test whether **3e** binds to the K⁺-bound IFS. The only state in our hands that binds K⁺ with sufficient affinity to yield high occupancy in cryo-EM maps is the intermediate iOFS state, and we reported its structure at 2.4 Å (DOI: 10.1038/s41467-023-38120-5). The map shows a very strong density for bound K⁺, consistent with the site proposed in the previous modeling (DOI: 10.1098/rstb.2008.0246) and crystallographic studies (DOI: 10.7554/eLife.02283 and DOI: 10.15252/embj.2021108341). ***In neither our previously published datasets (DOI: 10.1038/s41467-023-38120-5) nor the new dataset collected in this work (please see below) have we observed excess density at the K⁺ sites proposed in DOI: 10.1085/jgp.202012577 and DOI: 10.15252/embj.2019101468.*** It is possible that different ion-bound states exist and are functionally significant but are too transient to be captured by cryo-EM.

We agree with the reviewer that it would be interesting and important to know whether **3e** can bind to the K⁺-bound state. To test this, we have collected a new cryo-EM dataset on the EAAT3 double cysteine mutant constrained by cross-linking in a mixture of OFS and iOFS (DOI: 10.1038/s41467-023-38120-5 and DOI: 10.1073/pnas.2501627122) in the presence of 100 μM **3e** and 300 mM K⁺. We observed, as in an earlier study (DOI: 10.1038/s41467-023-38120-5), a mixture of structural classes corresponding to the K⁺-bound iOFS and apo OFS. Neither contained a clear density that could be attributed to **3e**, suggesting that the compound does not bind tightly to these states (**Response Fig. 2**).

We include these new data as supplementary figures (new **Appendix Fig. S3**) and discuss them in the context of DOI: 10.1085/jgp.202012577 and DOI: 10.15252/embj.

2. I am missing a more detailed description of the binding mechanism. The authors use only indirect approaches, such as thermal shift binding assays or impedance assay, which do not provide important information about the binding process and the transporters state of binding. For example, the authors suggest that "3e either selectively binds to apo SLC1A1g or displaces bound Na⁺"; I would insist that the authors distinguish between these two options with experimental approach. Is binding of 3e reduced at higher Na⁺ concentrations?

We share the Reviewer's puzzlement. In our *in vitro* thermal-shift binding experiments in detergent solution, there is no way to determine the order of events: whether **3e** binds to the apo transporter and prevents subsequent binding of Na⁺ ions, or binds to the Na⁺-bound transporter and displaces Na⁺ ions. However, we propose that, thermodynamically, Na⁺ and **3e** compete for binding to the transporter, including the coordinating residue T102. In the original submission, we reported that **3e** binds in the absence of Na⁺ in the thermal-shift assay, and that 200 mM Na inhibits this binding. Using the NanoDSF approach, we now also show that **3e** inhibits Na⁺ binding in a dose-dependent manner (**Response Fig. 1**).

3. Moreover, what is the evidence that the compound binds from the cytoplasm? At present, the authors only provide a hypothetical binding process, diffusion of 3e across the membrane and subsequent binding from the insight, I would urge the authors to support this or another binding path with experiments. Can the authors exclude that EAAT3 binds 3e from the outside and translocates to IF?

In cells, we envision that the hydrophobic **3e** and analogues partition into the membrane and bind to EAAT3 from within the bilayer after it releases the substrate and Na⁺ ions into the cytoplasm in IFS. We tested whether the differences in extracellular Na⁺ affect **3e** potency in cellular uptake. To this end, we measured the dose dependence of **3e** inhibition of radiolabeled L-Asp uptake into HEK293 cells expressing EAAT3 in the presence of saturating and near-Km concentrations of Na⁺ (**Response Fig. 3**). We observed **3e** inhibited L-[³H]-Asp uptake with values of 4.3 ± 0.5 μM at 100 mM Na⁺ and 1.8 ± 0.7 μM at 20 mM Na⁺, respectively.

Similar IC₅₀s suggest that extracellular Na⁺ does not significantly affect potency, consistent with intracellular Na⁺ being the determining factor and with **3e** targeting the IFS. Our new structures also show that the **3e** does not bind to OFS states (**Response Fig. 2**).

One could learn a lot with cell electrophysiological experiments to investigate the 3e block.

We fully agree with the reviewer. Looking ahead, we plan to establish e-phys assays to analyze new inhibitors and will include **3e** in these studies. However, we feel that they are beyond the scope of the current paper. However, to address the reviewer's concerns, we optimized a FLIPR-based assay using a membrane potential dye that enables determination of changes in membrane potential triggered by SLC1A1-dependent L-Glu uptake. These studies have confirmed the overall accuracy of our earlier impedance assays and offered a new (high-throughput amenable) assay system for future studies (please see new **Figs. 6E, 6F** and **Appendix Fig. S7G**)

Minor concerns

4. What is the evidence that **3e** indeed blocks the transporter? How does **3e** modify the EAAT3 anion channel? Can the authors provide on and off rates of binding?

The evidence for **3e** blocking transport is extensive, as it was identified in the fluorescence-based screen measuring cellular glutamate uptake. **3e** inhibits uptake in several assays that we have established, including in direct L-[³H]-Asp uptake assays. The **3e**-bound structure also features a wide-open HP2 gate, which would prevent transmembrane movement of the transport domain and thus blocks the transporter. Although our data show that EAAT3 is blocked by **3e** binding, measuring on- and off-rates is technically challenging due to the hydrophobic nature of **3e** and the complex interactions among the protein, compound, and lipid bilayer.

Whether the **3e** blocks the anion channel is unclear; we would need to perform electrophysiological experiments to show this explicitly. Notably, however, chloride channel opening requires sodium and glutamate binding. Since these are prevented by **3e**, the compound likely inhibits chloride conductance.

5. I remember from my Renal Physiology class that EAAT3 is exclusively expressed in the proximal tubules of the nephron. I am therefore surprised that it plays such a prominent role in the most common form of adult kidney cancer, and I have to admit that I kept thinking about the general role of EAAT3 in kidney during reading the whole manuscript. Does it mean that Renal Cell Carcinomas originate from the proximal tubule? Could the authors provide some explanation why a proximal tubule transport plays a role in renal cancer and speculate whether it is also expressed in other forms of cancers?

The reviewer is correct in noting that the ccRCCs originate from proximal tubules. We have edited the introduction to describe this information clearly (please see **lines 54-55**).

6. Moreover, since EMBO J is not a clinical journal and the prospective audience will mostly not consist of oncologists, I would suggest to give a better introduction into the whole idea, i.e. can we expect that an EAAT3 can cure renal cancer and what about other forms of cancer (see comment 5.)?

We thank the reviewer for this note and have edited the introduction accordingly (please see **lines 52-67**).

7. I find it concerning that the scientific basis for this study, i.e. that EAAT3 plays a role in kidney cancer has not been accepted for publication, but has only posted on biorxiv.

We are pleased to inform the reviewer that this manuscript was accepted in Nature Communications and is now available online (Grubb T et al. 2025).

Referee #2:

Renal Cell Carcinomas (RCCs) depend metabolically on the trimeric sodium-coupled aspartate and glutamate transporter SLC1A1/EAAT3. However, pharmacologically targeting SLC1A1 is challenging because the amino acid residues interacting with the substrates and the co transported ions in the binding pocket of EAAT3 are identical those of the related EAAT1 and EAAT2 transporters, which are essential for function of the brain. Recently a compound, termed 3e was found which selectively inhibits EAAT3. To obtain more potent inhibitors, cryogenic electron microscopy (cryo-EM) structure of human SLC1A1/EAAT3 bound to compound 3e was determined. The structure reveals the transporter in an inward-facing apo state where 3e is located between the transport and scaffold domains and identifies a molecular determinant of the selectivity of 3e for EAAT3. Structure-guided modifications of the 3e scaffold led to the identification of two SLC1A1 selective inhibitors with improved cancer cell-killing potency and suggest a potential for the development of SLC1A1-targeted therapies. This is a well executed multi-disciplinary study, combining structural biology, synthetic organic chemistry and cell biology. Despite all the work involved, the two new inhibitors show only modest improvement of cancer cell-killing. However, the study provides some new mechanistic insights, such as the location of the inhibitor site which partly overlaps with the Na³ site at T102 as well as the identification of F99 as an important determinant of selectivity.

We thank the Reviewer for appreciating the significance of the work. We agree that the relatively modest improvement in potency may cause challenges in this developing this series any further; however, our work has now explained the key problems with these molecular scaffolds. The BIAs are highly hydrophobic and engage SLC1A1 in a membrane-embedded pocket. Consequently, it is difficult to improve their solubility without losing on-target potency. In line with the reviewer's sentiment, we see the major value of work being: (1) the development of a new and rigorous structure-guided medicinal chemistry pipeline for future target discovery; (2) new mechanistic insights into the SLC1A1 transport cycle and the discovery of a previously unrecognized druggable 'cryptic' pocket; and (3) the physiological relevance and feasibility of targeting SLC1A1 as a complementary approach to existing therapies in kidney cancer, including HIF2 α blockade.

The Authors use impedance as a read-out of transport and show that 3e and the new compounds selectively inhibit EAAT3. However, impedance measurements are not very familiar to most readers, including those in the transporter field. This technique should be clearly described, indicating its advantages and drawbacks relative to the established methodology. What the Authors show as a cartoon in Fig. 6A without any further explanation is clearly not sufficient.

We apologize for this oversight and have expanded the description (lines 350-353) as suggested.

In addition, conventional transport measurements such as radioactive uptake should be done in parallel with some of the impedance measurements to underscore its validity as readout of transport.

We have performed the experiments as suggested (**Response Fig. 3**). In addition, we have also performed a colorimetric assay to measure changes in intracellular L-Asp in the presence or absence of the SLC1A1 inhibitors. These data are presented in new **Fig. 7C**. Together, these new studies demonstrate that the SLC1A1 inhibitors block L-Asp uptake and lead to reduced intracellular L-Asp pools.

Moreover, it is important to determine kinetic parameters to establish the mechanism of inhibition by the compounds.

Establishing the exact mechanism of inhibition in this system may be challenging, given the complex interplay among the inhibitor, ions, and membrane components. We have shown in the initial submission and further expanded now (**Response Fig. 1**) that **3e** and Na⁺ compete for binding to the transporter. In contrast to competitive binding in detergent solutions, the IC₅₀ in cell-based assays is not strongly affected by extracellular Na⁺, consistent with a mechanism in which **3e** diffuses through the membrane and binds to the apo IFS from the bilayer. Its binding on the domain interface together with cholesterol suggests that it would also impede transport domain movements. Finally, we have now begun optimizing a FLIPR assay to measure SLC1A1 transport. We will deploy this approach in combination with changes in order of addition (e.g., substrate then inhibitor versus inhibitor then substrate) in future experiments to obtain more granular kinetic details of novel inhibitors.

A minor point is the mentioning of F99 as "the determinant" of selectivity (line 350). To support such a statement, phenylalanine should be introduced at the corresponding positions of EAAT1 and EAAT2 to see if now these transporters become sensitive to **3e**. That would be great, but without such proof one should say that F99 is "a major determinant" of selectivity.

We thank the reviewer for this note and have updated our text to indicate either that 'F99 is a key determinant' or that 'F99 is a major determinant' of selectivity (please see lines **455-456** and **113-115**).

In addition, because of the proximity of the inhibitor to the Na³ site, it would be worthwhile to check if the inhibition of transport by **3e** is increased at lower sodium concentrations.

We thank the reviewer for this comment and have addressed this concern experimentally. This data is provided in **Response Fig. 3** and **Appendix Fig. S2D**.

Dear Abhishek,

We have now received re-review reports from two referees, which I have included below. As you will see, you have addressed their concerns satisfactorily. However, before I can finally accept the manuscript, there are some remaining editorial points which need to be addressed. In this regard would you please:

- rename the conflict of interest statement a "Disclosure and Competing Interests Statement",
- include an OrcidID for Shaun Stauffer,
- remove the AC/CrediT section from the text,
- acknowledge funding from grant DoD-KCRP-IDA (HT9425-25-1-0761) and Shared Laboratory Services at the Lerner Research Institute in the manuscript and our online submission system,
- use the following section order: Title page - Abstract & Keywords - Introduction - Results - Discussion - Methods - Data Availability - Acknowledgments - Disclosure Statement & Competing Interests - References - Figure Legends - (Main Tables with legends if applicable) - Expanded View Figure Legends,
- instead of using "Supplementary" (Materials), use "S" as nomenclature; Appendix tables need to be referred to as Appendix Table S1, etc. (also in the Tools and Reagents table),
- save the appendix file as a pdf; this file should have a title page reading "Appendix for Mechanism and Structure-Guided Optimization of SLC1A1/EAAT3-Selective Inhibitors in Kidney Cancer" followed by a table of contents listing each item and its page number (without author list and affiliations),
- upload completed Source Data checklists,
- ensure uploaded data sets are fully available to the public,
- provide exact p values in the legends of figures 3D, E; 5E, F-H; 7C-F, G, H, state the statistical test used for data analysis in the legends of figures 3E, 7G, H,
- define box plots in terms of minima, maxima, centre, bounds of box and whiskers, and percentile in the legend of figure 4A,
- define 'n' in the legends of figures 1F, G; 3D, 5E, F-H; 6C, F; 7G, H, and
- define error bars in the legends of figures 3D, 6C, F; 7G, H

We include a synopsis of the paper (see <http://emboj.embojpress.org/>). Please provide me with a general summary image, a two sentence statement and 3-5 bullet points that capture the key findings of the paper.

I am looking forward to receiving your revised manuscript.

EMBO Press is an editorially independent publishing platform for the development of EMBO scientific publications.

Best wishes,

William

William Teale, PhD
Editor
The EMBO Journal
w.teale@embojournal.org

Read our guidance for manuscript revisions and related editorial policies: <https://link.springer.com/journal/44318/submission-guidelines#cms-Revised-submissions>

<https://media.springernature.com/original/springer-cms/rest/v1/content/27825798/data/v1>

- a point-by-point response to the referees' comments, with a detailed description of the changes made (as a word file).
- a word file of the manuscript text.
- individual production quality figure files (one file per figure)
- a complete author checklist
- Expanded View files (replacing Supplementary Information)
- a Reagents and Tools Table as part of the Methods section

Please remember: Digital image enhancement is acceptable practice, as long as it accurately represents the original data and conforms to community standards. If a figure has been subjected to significant electronic manipulation, this must be noted in the figure legend or in the 'Methods' section. The editors reserve the right to request original versions of figures and the original images that were used to assemble the figure.

We realize that it is difficult to revise to a specific deadline. In the interest of protecting the conceptual advance provided by the work, we recommend a revision within 3 months (25th May 2026). Please discuss the revision progress ahead of this time with the editor if you require more time to complete the revisions. Use the link below to submit your revision:

Referee #1:

This manuscript by Koochaki reports the development of novel treatment options for Renal Cell Carcinomas (RCCs). Based on earlier work of the Chakraborty group that upregulation of SLC1A1/EAAT3 glutamate transporter is associated with this disease the authors combine cell biology, structural biology and medical chemistry to improve an existing EAAT3 blocker (named 3e) that may help future cancer treatment in certain patients.

This is a very important project; the use of high profile basic research to answer pressing medical questions is very appealing, and the combination of multiple approaches is very impressive.

The authors have satisfactorily addressed all my concerns, and I have no further criticisms. I recommend acceptance in its present form.

Referee #2:

The Authors have adequately dealt with my concerns (see my original review).

Dear Dr. Teale,

We thank you and the reviewers for the encouraging remarks about our manuscript and are delighted that it has been deemed suitable for publication. The reviewers did not have any additional concerns; however, we are responding to the editorial concerns that were raised in the decision letter.

1. There were some similarities between image panels in Appendix Figure S1B and S3B; I know that this is the type of image that confuses image analysis software, but could you check that you have indeed selected different individual proteins for those pictures shown?
Response: We can confirm that the image panels in Appendix Figures S1B and S3B come from two different datasets. They are selected 2D class averages of hEAAT3g (Figure S1B) and EAAT3-X (Figure S3B), respectively. Although the two proteins adopt inward-facing and outward-facing states, their 2D projections appear similar.
2. Rename the conflict of interest statement a "Disclosure and Competing Interests Statement"
Response: Done
3. Include an OrcidID for Shaun Stauffer
Response: Done
4. Remove the AC/CrediT section from the text
Response: Done
5. Acknowledge funding from grant DoD-KCRP-IDA (HT9425-25-1-0761) and Shared Laboratory Services at the Lerner Research Institute in the manuscript and our online submission system
Response: Done
6. Use the following section order: Title page - Abstract & Keywords - Introduction - Results - Discussion - Methods - Data Availability - Acknowledgments - Disclosure Statement & Competing Interests - References - Figure Legends - (Main Tables with legends if applicable) - Expanded View Figure Legends
Response: Done
7. Instead of using "Supplementary" (Materials), use "S" as nomenclature; Appendix tables need to be referred to as Appendix Table S1, etc. (also in the Tools and Reagents table)
Response: Done
8. Save the appendix file as a pdf; this file should have a title page reading "Appendix for Mechanism and Structure-Guided Optimization of SLC1A1/EAAT3-

Selective Inhibitors in Kidney Cancer" followed by a table of contents listing each item and its page number (without author list and affiliations)

Response: Done

9. Upload completed Source Data checklists

Response: Done

10. Ensure uploaded data sets are fully available to the public

Response: Done

11. Provide exact p values in the legends of figures 3D, E; 5E, F-H; 7C-F, G, H

Response: Done. Please note that we used the p-values calculated in prism. Unfortunately, the software adjusted everything below 0.0001 to just <0.0001. We have provided exact values of everything ≥ 0.0001 and used the # to mark all p-values at or below this threshold.

12. State the statistical test used for data analysis in the legends of figures 3E, 7G, H

Response: Done

13. Define box plots in terms of minima, maxima, centre, bounds of box and whiskers, and percentile in the legend of figure 4A

Response: Done

14. Define 'n' in the legends of figures 1F, G; 3D, 5E, F-H; 6C, F; 7G, H

Response: Done

15. Define error bars in the legends of figures 3D, 6C, F; 7G, H

Response: Done

16. Synopsis Description:

Dysregulation of the SLC1A1/EAAT3 L-Asp/L-Glu transporter causes neurological disorders and cancer, establishing the need to develop selective pharmacological agents to target this protein. Using a structure-guided medicinal chemistry campaign and cell-based validations, we report the discovery of a membrane-embedded (hidden) pocket that can be targeted to block SLC1A1's transport cycle and pharmacologically exploited to block cancer cell growth.

Highlights:

- Cryo-EM structure of SLC1A1 with a recently reported bicyclic imidazopyridine amine (BIA) inhibitor bound in a 'cryptic' orthosteric pocket.
- The inhibitor shows a dual mechanism of inhibition: competitively blocking Na^+ (and consequent substrate) binding and wedging itself between the scaffold and transport domain to obstruct SLC1A1's transport cycle.
- Medicinal chemistry coupled with biophysical and cell-based assays identify two analogs with improved potency.

- Biochemical measurements and rescue experiments confirm that the BIA inhibitors are on-target and cause antiproliferative effects by blocking SLC1A1-dependent metabolic programs.

Dear Abhishek,

I am pleased to inform you that your manuscript has been accepted for publication in the EMBO Journal.

Congratulations to you and all involved!

You may qualify for financial assistance for your publication charges - either via a Springer Nature fully open access agreement or an EMBO initiative. Check your eligibility: <https://link.springer.com/journal/44318/how-to-publish-with-us>

Best wishes,

William

William Teale, PhD
Editor
The EMBO Journal
w.teale@embojournal.org

Please note that it is The EMBO Journal policy for the transcript of the editorial process (containing referee reports and your response letters) to be published as an online supplement to each paper. If you should prefer removal of any referee-only figures included in the point-by-point response(s), e.g. because they may still be used for future publication or because they have been reproduced from published work by others, please do let us know immediately via response email.

More information is available here: <https://link.springer.com/partners/embo-press/editorial-policies#Peer%20review>
